# Activity-dependent Golgi satellite formation in dendrites reshapes the neuronal surface glycoproteome

Anitha P Govind[1†], Okunola Jeyifous[1,2†], Theron A Russell[1], Zola Yi[1], Aubrey V Weigel[3], Abhijit Ramaprasad[1], Luke Newell[1], William Ramos[1], Fernando M Valbuena[4], Jason C Casler[4], Jing-Zhi Yan[5], Benjamin S Glick[4], Geoffrey T Swanson[5], Jennifer Lippincott-Schwartz[3]*, William N Green[1,2]*

[1]Department of Neurobiology, University of Chicago, Chicago, United States; [2]Marine Biological Laboratory, Woods Hole, United States; [3]Janelia Research Campus, Howard Hughes Medical Institute, Ashburn, United States; [4]Department of Molecular Genetics and Cell Biology, The University of Chicago, Chicago, United States; [5]Department of Pharmacology, Northwestern University, Feinberg School of Medicine, Chicago, United States

*For correspondence:
lippincottschwartzj@janelia.hhmi.org (JL-S);
wgreen@uchicago.edu (WNG)

†These authors contributed equally to this work

**ABSTRACT** Activity-driven changes in the neuronal surface glycoproteome are known to occur with synapse formation, plasticity, and related diseases, but their mechanistic basis and significance are unclear. Here, we observed that N-glycans on surface glycoproteins of dendrites shift from immature to mature forms containing sialic acid in response to increased neuronal activation. In exploring the basis of these N-glycosylation alterations, we discovered that they result from the growth and proliferation of Golgi satellites scattered throughout the dendrite. Golgi satellites that formed during neuronal excitation were in close association with endoplasmic reticulum (ER) exit sites and early endosomes and contained glycosylation machinery without the Golgi structural protein, GM130. They functioned as distal glycosylation stations in dendrites, terminally modifying sugars either on newly synthesized glycoproteins passing through the secretory pathway or on surface glycoproteins taken up from the endocytic pathway. These activities led to major changes in the dendritic surface of excited neurons, impacting binding and uptake of lectins, as well as causing functional changes in neurotransmitter receptors such as nicotinic acetylcholine receptors. Neural activity thus boosts the activity of the dendrite's satellite micro-secretory system by redistributing Golgi enzymes involved in glycan modifications into peripheral Golgi satellites. This remodeling of the neuronal surface has potential significance for synaptic plasticity, addiction, and disease.

## Introduction

How the surface glycoproteome of neurons is established and regulated during neural plasticity has been an enduring question in neuronal cell biology. In particular, where and when different classes of glycoproteins are synthesized, modified, and traverse from their birthplace to their functional sites remain dominant issues. It is known that decentralized protein synthesis in dendrites plays a significant role in protein targeting to remote sites, with integral membrane and secreted proteins locally processed and trafficked in dendrites through a micro-secretory system (*Kennedy and Hanus, 2019*). Within this system, newly synthesized proteins move from the endoplasmic reticulum (ER) to ER exit sites (ERESs) scattered across the dendritic ER membrane (*Aridor et al., 2004*). The proteins then appear to sidestep the perinuclear Golgi and instead are routed to local endosomal compartments before being delivered to the plasma membrane (*Hanus and Ehlers, 2008*; *Bowen et al., 2017*). This

so-called 'Golgi bypass' pathway conveys glycoproteins with predominantly immature sugar chains to the plasma membrane, giving rise to the atypical, 'high-mannose', glycosylation pattern characteristic of dendritic membrane surfaces (*Hanus et al., 2016*). A major advantage of the Golgi bypass pathway is that it allows locally synthesized proteins in dendrites to avoid having to undergo long distance trafficking to and from the perinuclear Golgi apparatus in the soma. Golgi bypass comes at a cost, however, since newly synthesized glycoproteins will no longer be processed by Golgi enzymes, which convert high-mannose sugars on glycoproteins to complex, sialyated forms to expand the glycoprotein's functional repertoire at the plasma membrane.

During neuronal excitation, the Golgi apparatus surprisingly undergoes dispersal from a single perinuclear structure into hundreds of scattered mini-Golgi structures within the soma of cultured neurons (*Thayer et al., 2013*). Neuronal excitation also significantly increases levels of sialic acid on *N*-linked glycoproteins at dendritic surfaces and their associated synapses (*Scott and Panin, 2014*; *Rutishauser, 2008*; *Boll et al., 2020*; *Torii et al., 2014*), with desialyation of glycoproteins in the brain leading to neuroinflammation, dysmyelination, and cognitive dysfunction (*Scott and Panin, 2014*; *Varki and Gagneux, 2012*). Because sialylation of glycoproteins occurs in the Golgi apparatus, we wondered whether the increased sialyation of glycoproteins occurring in excited or stimulated neurons was at all related to the phenomenon of Golgi dispersal seen during neuronal stimulation. In particular, we asked whether Golgi dispersal is causative in altering distal secretory and endocytic trafficking pathways to enable the dendritic neuronal surface glycoproteome to be sialyated for optimal functioning in response to excitation.

To test this possibility, we set up imaging and biochemical experiments for studying the effects of membrane excitability on Golgi distribution and function in either non-neuronal cells or neuronal cultures, examining how the Golgi disperses in response to membrane excitability. We specifically addressed whether Golgi dispersal in neurons impacts the distribution of sialyltransferases and other glycan modifying enzymes, and whether surface receptor glycosylation patterns, neurotransmitter receptor function, and/or the binding and uptake of lectins are altered as a result. We found that membrane excitation in either neurons or non-neuronal cells led to dissolution of the Golgi's compact, centralized appearance, and to the formation of scattered Golgi elements that were adjacent to ERESs as well as endosomes. The disseminated Golgi elements, representing Golgi satellites in dendrites, functioned as part of a micro-secretory/glycosylation system, in which *N*-glycans attached to either newly synthesized receptors or endocytosed surface receptors were remodeled into complex, sialylated forms. The Golgi-mediated sialyation of glycoproteins had cellular consequences, including increased uptake of the lectin wheat germ agglutinin (WGA) and functional upregulation of specific neurotransmitter receptors. Under neuronal excitation, therefore, the development of Golgi satellites and their associated trafficking pathways could help explain how the neuronal surface glycoproteome is modified, and may be relevant to mechanisms underlying synaptic plasticity, addiction, and disease.

## Results

### Golgi dispersal by nicotine occurs in non-neuronal cells expressing α4β2 receptors

To study the process of excitability-induced Golgi fragmentation, we developed a live cell microscopy assay for examining Golgi dispersal in response to changes in membrane excitability induced by nicotine. In the brain, α4β2Rs are the main class of nicotinic receptors with high affinity for nicotine (*Albuquerque et al., 2009*). In our assay, we expressed α4β2R subunits in non-excitable human embryonic kidney (HEK293) cells (*Vallejo et al., 2005*) to make these cells nicotine-excitable and then examined the effect nicotine treatment had on Golgi morphology.

When α4β2R-expressing HEK293 cells were exposed to 10 µM nicotine, the Golgi dramatically changed from having a compact perinuclear localization to being distributed among discrete smaller elements dispersed throughout the cytoplasm, as observed by localizing exogenously expressed GFP-tagged β-1,4,galactosyltransferase (GalT-GFP), a late-acting Golgi enzyme (*Figure 1A*, +nicotine). The changes in Golgi morphology were initiated within 1 hr of nicotine exposure, lasted as long as nicotine was present, and were reversible with nicotine removal (see *Figure 1—figure supplement 1*). Untreated cells imaged over the same time period showed no Golgi fragmentation, indicating fragmentation was induced by nicotine treatment (*Figure 1A*, -nicotine).

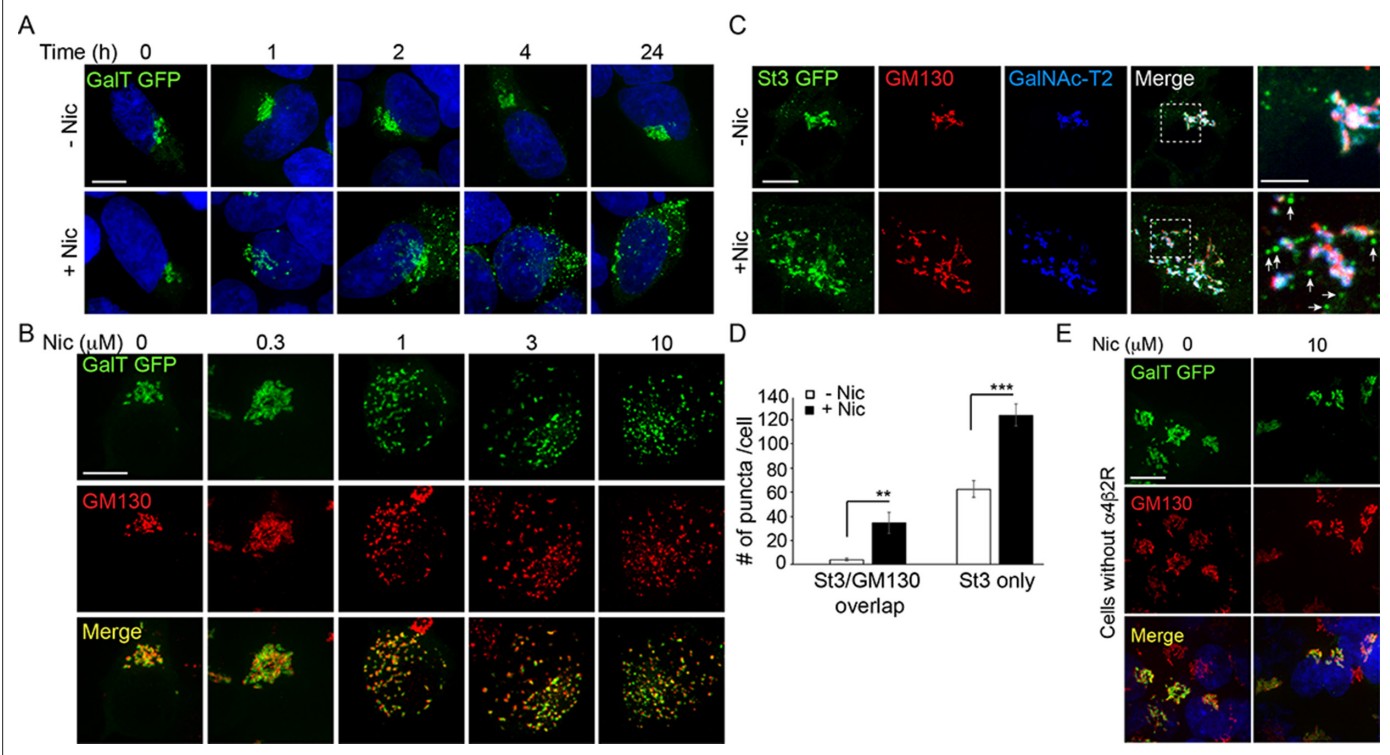

**Figure 1.** Golgi dispersal by nicotine exposure in non-excitable, α4β2R-expressing human embryonic kidney (HEK) cells. (**A**) Time course of Golgi dispersal by nicotine. HEK293 cells stably expressing α4β2Rs (α4β2R cells) were transfected with GFP-tagged galactosyltransferase (GalT, green) and imaged at the indicated times after nicotine exposure (10 µM). Cells were also labeled with DAPI (blue) after fixation. Scale bar, 10 µm. (**B**) Dose dependence of Golgi dispersal by nicotine. Cells were fixed after 17 hr of nicotine exposure at the indicated concentrations and then stained with anti-GM130 antibody (red) and DAPI (blue) before imaging. Scale bar, 10 µm. (**C**) Heterogeneity of Golgi components in nicotine-dispersed Golgi fragments. α4β2R cells were transfected with GFP-GalT (blue) and St3-GFP (green) and nicotine-treated as in A prior to fixation and staining for GM130 (red). Scale bar, 10 µm. (**D**) Numbers of dispersed Golgi puncta per cell that contain both GM130 and St3, or only St3. Total number of puncta per cell, data are shown as mean ± SEM. For left graph, untreated cells, 3.9 ± 1.1; nicotine-treated cells, 34.6 ± 9.0 (n = 8–10 cells per group, **p < 0.05). For right graph, control cells, 62.6 ± 6.9; nicotine cells, 124.1 ± 8.9 (n = 10 cells per group, ***p < 0.00001). (**E**) Nicotine-treatment does not alter Golgi morphology in cells that do not express α4β2R. HEK293 cells without α4β2Rs were transfected with GalT-GFP and then treated with or without nicotine (10 µM) for 17 hr and imaged. Scale bar, 10 µm.

The online version of this article includes the following figure supplement(s) for figure 1:

**Figure supplement 1.** Reversal of nicotine-induced Golgi dispersal.

**Figure supplement 2.** Classification and quantification of cells into three categories (intact, partially dispersed, and fully dispersed) based on Golgi integrity and morphology.

Golgi dispersal was initiated at nicotine concentrations as low as 300 nM and saturated at ~1 µM nicotine, as shown in cells expressing GalT-GFP and antibody labeling for the Golgi structural marker, GM130 (**Figure 1B**). The dose dependence for Golgi fragmentation was similar to the dose dependence for nicotine sensitivity of neurons (**Vallejo et al., 2005**; **Govind et al., 2012**). Golgi dispersal by nicotine varied somewhat among α4β2R-expressing cells (**Figure 1—figure supplement 2B**), most likely due to cell-to-cell differences in α4β2R expression, a feature that increases with cell passage number.

To examine whether different Golgi components redistributed equivalently into Golgi puncta during nicotine treatment, we performed triple-labeling experiments using markers for GM130, sialyltransferase (St3-GFP), and/or galactosyltransferase 2 (GalNAc-T2)(**Figure 1C**). GM130 is a scaffolding protein peripherally associated with the Golgi and is involved in microtubule organization at the Golgi apparatus (**Wu and Akhmanova, 2017**). St3 and GalNAc-T2 are both transmembrane Golgi enzymes involved in glycoprotein modifications. We found that some Golgi puncta contained all three Golgi markers, whereas others had only GalNAc-T2 or St3-GFP associated with them and no GM130 labeling (**Figure 1C**, see arrows in zoomed boxes). The numbers of these puncta were significantly increased

by nicotine exposure (*Figure 1D*). No change in Golgi morphology was observed in HEK293 cells not expressing α4β2Rs after overnight nicotine treatment (*Figure 1E*), indicating α4β2R expression was required for nicotine-dependent Golgi dispersal.

## Nicotine and other excitatory stimuli trigger Golgi dispersal within somata and dendrites of cultured neurons

In neurons, the Golgi is primarily localized in the soma, but small, distinct populations of scattered Golgi elements are also present. One population of scattered Golgi elements are called 'Golgi outposts' (*Horton and Ehlers, 2003*). These structures are enriched in GM130 and other Golgi enzymes, and usually localize only to the most proximal part of the apical dendrite or at dendrite branch points (*Horton and Ehlers, 2003*). Another population of scattered Golgi elements are called 'Golgi satellites' (*Mikhaylova et al., 2016*). These structures are usually smaller than Golgi outposts and are widely dispersed in dendrites. Golgi satellites lack GM130 and contain glycosylation machinery (*Mikhaylova et al., 2016*). They have been shown to be near ERESs as well as recycling endosomes, and thus could contribute to the micro-secretory system occurring in dendrites (*Mikhaylova et al., 2016*). Given this prior work, we tested whether nicotine exposure to α4β2R-expressing neurons led to Golgi dispersal, and if so, whether this led Golgi enzymes to redistribute to dendritic Golgi outposts and/or Golgi satellites, increasing the size and/or abundance of these structures.

We used cultured neurons from rat cortex (unless otherwise specified) to study the effect of nicotine exposure on Golgi localization. Within these cultures, 5–10% of neurons are endogenously expressing α4β2Rs, with their expression largely found in inhibitory neurons (*Govind et al., 2012*). To ensure the neurons being examined in this system were expressing α4β2Rs and responsive to nicotine, we transfected the neurons with α4 and HA-tagged β2 subunits. We further expressed fluorescent protein (FP)-tagged Golgi enzymes or stained with marker-specific antibodies in order to examine the Golgi in these cells.

Neurons expressing α4β2Rs and FP-tagged Golgi markers for St3-GFP, GM130, GalNAc-T2, and/or mannosidase II (Man II) were treated with or without nicotine and the distributions of the Golgi markers were examined (*Figure 2A–C*). Examining the cell body of these neurons revealed that nicotine treatment caused Golgi structures to fragment into scattered elements (*Figure 2Ai–Ci*; see boxed areas in images labeled with the different Golgi markers and their corresponding enlarged images). Similar to Golgi dispersal induced by nicotine in non-neuronal cells expressing α4β2Rs, both small and large Golgi puncta appeared over time during nicotine treatment. Large Golgi puncta contained all of the different Golgi markers, whereas small Golgi structures contained mainly Golgi enzymes (i.e., St3, GalNAc-T2, and Man II) without GM130 (*Figure 2Ai–Ci*).

We next turned our attention to nicotine's effect on Golgi-related organelles in the dendrites of these neurons (i.e., Golgi outposts and satellites) (*Figure 2A–C*; boxed areas in inverted images labeled with [ii] and their corresponding enlarged images). In the absence of nicotine, GM130-labeled Golgi outposts near the apical dendritic zone were present, as were numerous Golgi satellite structures dispersed throughout the dendrite that contained St3, GalNAc-T2, and/or Man II, but lacked GM130 (*Figure 2Aii–Cii*). Nicotine treatment did not change the low number of GM130-positive Golgi outposts; however, the GM130-negative Golgi satellite structures containing GalNAc-T2, St3, and/or Man II underwent a significant increase in number in response to nicotine treatment (*Figure 2Aii–Cii*). The increase in Golgi satellites in dendrites during nicotine treatment was also observed in untransfected neurons stained with antibodies to St3 (*Figure 2D–E*). Additional images are displayed in *Figure 2—figure supplements 1 and 2*, which tested the specificity of St3 antibody staining in neurons.

To determine whether stimulating neurons by means other than nicotine treatment could also increase Golgi satellite abundance in dendrites, we transfected neurons with St3-Halo to label Golgi satellites and fluorescent Venus to fill the cytoplasm for dendrite identification. The synaptic activity of cortical cultures was then increased by applying either the GABA$_A$ receptor antagonist bicuculline for 18–20 hr or by withdrawing the NMDA receptor antagonist APV for 18–20 hr following 2 days of APV treatment (*Figure 2F and G*; *Thayer et al., 2013*). An increase in the number and density of St3-containing puncta per dendrite length was observed for both treatments compared to vehicle control or chronic APV treatment (*Figure 2F–H*). We thus concluded that changes in neuronal activity alone

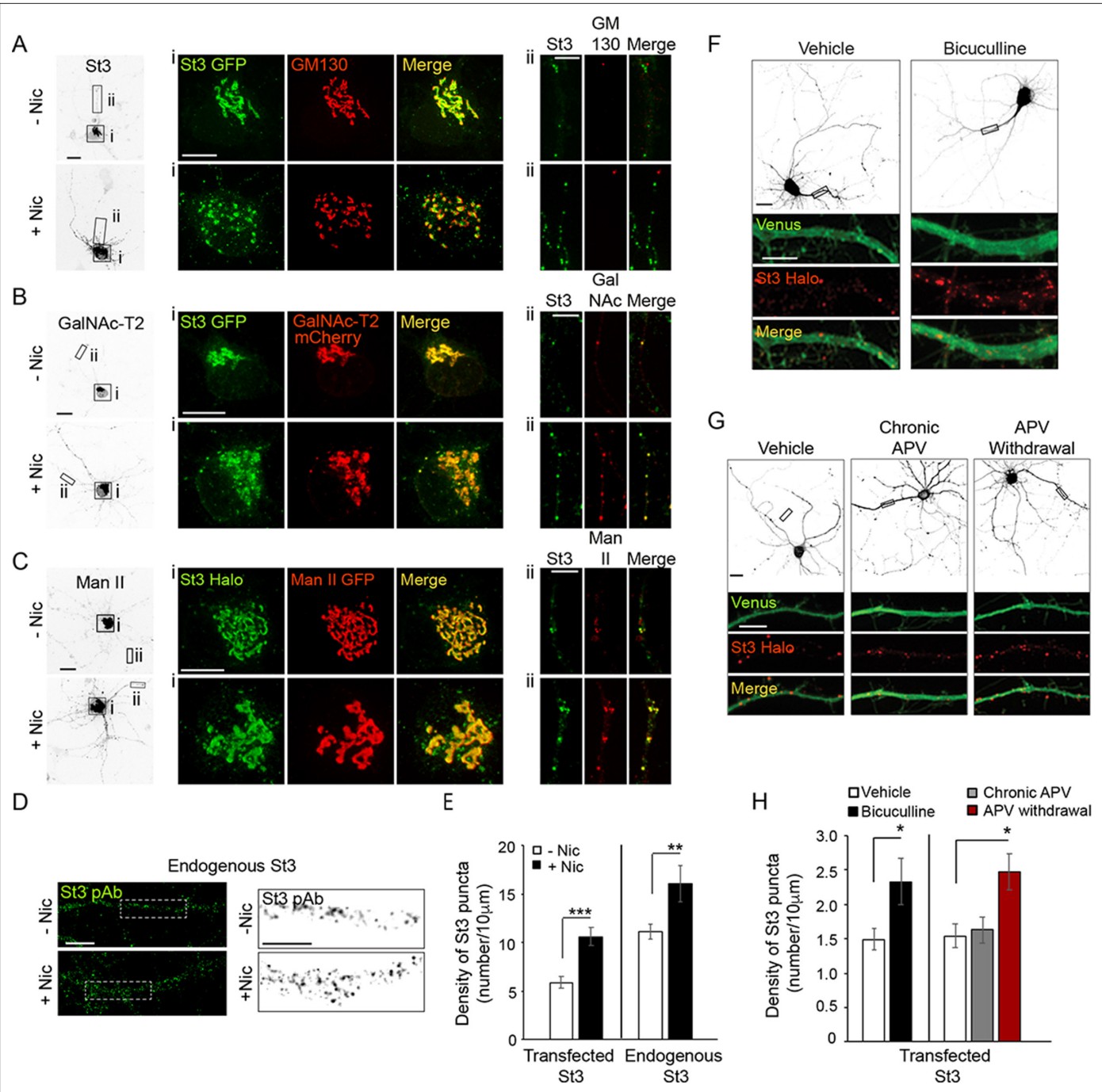

**Figure 2.** Nicotine- and activity-dependent Golgi dispersal and Golgi satellite formation in cultured neurons. (**A–E**) Nicotine-dependent changes. Scale bars, 10 μm. (**A**) Neurons were treated with or without nicotine, and the Golgi was imaged with St3 and GM130 markers. DIV10 cortical neurons (E18 rat pups) were transfected with St3-GFP (green) and HA-tagged α4β2R subunits. Cultures were fixed and stained for GM130 (red). Low magnified images of neurons treated with (bottom, +Nic) or without (top, -Nic) 1 μM nicotine for 17 hr. (i) Images of St3- and GM130-containing Golgi in somata. (ii) Images of St3- and GM130-containing Golgi in dendrites. (**B**) Neurons were treated with or without nicotine, and the Golgi was imaged using St3 and GalN as markers. Cultures were transfected, nicotine-treated or untreated, and analyzed as in A, with the additional transfection of GalN-mCherry (red) replacing the GM130 marker. (i) Images of St3- and GalN-containing Golgi in the soma. (ii) Images of St3- and GalNAc-T2-containing Golgi in dendrites. (**C**) Neurons were treated with or without nicotine and the Golgi was imaged using St3 and Man markers. Cultures were transfected, nicotine-treated or untreated, and analyzed as in B, except with transfection of Man II GFP (Man-GFP; red) replacing GalNAc-T2-mCherry. (i) Images of St3- and Man II-containing Golgi in the somata ii. Images of St3- and Man II-containing Golgi in dendrites. (**D**) Images of nicotine-treated (+Nic) and untreated (-Nic) dendrites, with endogenous St3 labeled with polyclonal anti-St3 antibody. Boxed regions (left panels) are displayed as magnified grayscale images in right panels. (**E**) Quantification of both endogenous St3 puncta density from D and transfected St3 puncta density from C plotted as number of

*Figure 2 continued on next page*

*Figure 2 continued*

St3 puncta per 10 µm. Data are displayed as mean ± SEM for transfected St3 in untreated cells (5.9 ± 0.6) or for nicotine-treated cells (10.6 ± 1.0) (n = 9 neurons and 18 dendrites per group, ***p < 0.000001); and for endogenous St3 in untreated cells (11.1 ± 0.8) or nicotine-treated cells (16.1 ± 1.9) (n = 8–11 neurons per group, **p < 0.008). (**F–H**) Activity-dependent changes. (**F**) Neurons transfected with St3-Halo and Venus were treated with bicuculline to increase synaptic activity and Golgi structures (St3-Halo) and cytoplasm (Venus) imaged. Low magnified images of neurons treated with (right) or without (vehicle, left) 20 µM bicuculline for 17 hr. Scale bar, 20 µm. Below are images of corresponding dendrites (black rectangles above) transfected with St3-Halo and Venus. Scale bar, 5 µm. (**G**) Neurons transfected with St3-Halo and Venus were treated first with 200 µM APV for 2 days to block NMDA-type glutamate receptors and then the APV was either continued (chronic APV) or withdrawn for 1 day to increase synaptic activity (APV withdrawal). The Golgi was then imaged using St3-Halo. Low magnified images show untreated neurons (vehicle), neurons treated chronically with APV, and neurons having APV withdrawed. Scale bar, 20 µm. The boxed regions are shown as enlarged images below this, and show the increase in St3-Halo labeled puncta in response to APV withdrawal. Scale bar, 5 µm. (**H**) Quantification of data from F and G measuring number of St3 puncta in dendrites per 10 µm. Data are displayed as mean ± SEM. Left: untreated cells, 1.5 ± 0.2; bicuculline-treated cells, 2.3 ± 0.3 (n = 12 neurons per group, *p < 0.04). Right: untreated cells, 1.5 ± 0.2; chronic APV cells, 1.6 ± 0.2; APV withdrawal cells, 2.5 ± 0.3 (n = 7 neurons per group, *p < 0.03). Quantitative analysis was conducted on two independent culture preparations.

The online version of this article includes the following figure supplement(s) for figure 2:

**Figure supplement 1.** Specificity of the St3 polyclonal antibody (pAb).

**Figure supplement 2.** Bicuculline treatment of non-transfected neuronal cultures resulted in the same increase in the number of St3 antibody-stained puncta as nicotine treatment, which required exogenous expression of α4β2Rs to observe the increase.

---

are sufficient to drive the formation of enzyme-rich Golgi satellites in soma and dendritic processes of neurons.

## Newly synthesized proteins traffic through dispersed Golgi satellites to reach the plasma membrane in stimulated neurons

We next investigated whether newly formed Golgi satellites arising with neuronal excitation participate in secretory trafficking of newly synthesized cargo. To address this question, we assayed the trafficking itinerary of secretory cargo in nicotine-treated cells using the retention using selective hooks (RUSH) system (*Boncompain et al., 2012*). The RUSH system enables controlled release of newly synthesized ER proteins from the ER into the secretory pathway by breaking a bond between an ER-resident protein and a cargo protein through biotin addition. As regulatable cargo in this system, we used fluorescent mApple-tagged GPI bound to the ER-resident protein KDEL-streptavidin (i.e., RUSH-mApple-GPI).

We introduced RUSH-mApple-GPI into non-neuronal HEK293 cells co-expressing α4β2Rs and St3-GFP and treated these cells with nicotine to fragment the Golgi. RUSH-mApple-GPI was then released from the ER through biotin addition and followed over time (*Figure 3A*). RUSH-mApple-GPI could be seen quickly passing through St3-GFP-labeled Golgi fragments before arriving at the plasma membrane in time lapse sequences. The kinetics of this trafficking resembled that for RUSH-mApple-GPI in cells not treated with nicotine (*Figure 3—figure supplement 1*). Thus, dispersed Golgi elements in nicotine-treated cells behave as bona fide units of the secretory pathway.

Because of technical difficulties in applying the RUSH system to neurons (i.e., there was some 'leakage' from ER prior to release of the ER-bound cargo), we used a different system to release bound fluorescent cargo from ER in order to test the role of Golgi satellites in secretory trafficking in neurons. The system used a modified version of a regulatable fluorescent secretory cargo called ESCargo (*Casler et al., 2020*). ESCargo forms homo-aggregates in the ER lumen when expressed but can be released into the secretory pathway upon addition of a membrane-permeable ligand that dissociates the aggregates (*Casler et al., 2020*). In neurons expressing ESCargo and Man-GFP, ER aggregates in the soma, dendrites, and axons could be seen prior to release in nicotine-treated cells (*Figure 3B*). Upon addition of the membrane-permeable ligand, the aggregates of ESCargo disassembled, resulting in soluble ESCargo rapidly filling the ER lumen due to its increased ability to diffuse (*Figure 3C*). Thereafter, the cargo moved out of the ER into non-motile Man II-containing Golgi puncta distributed along the dendrites. The intensity of ESCargo in many Golgi puncta could be seen to subsequently diminish, consistent with the cargo's downstream trafficking and delivery to the plasma membrane. These data provide evidence that newly synthesized proteins traffic from the ER through dispersed Golgi satellites en route to the plasma membrane in stimulated neurons.

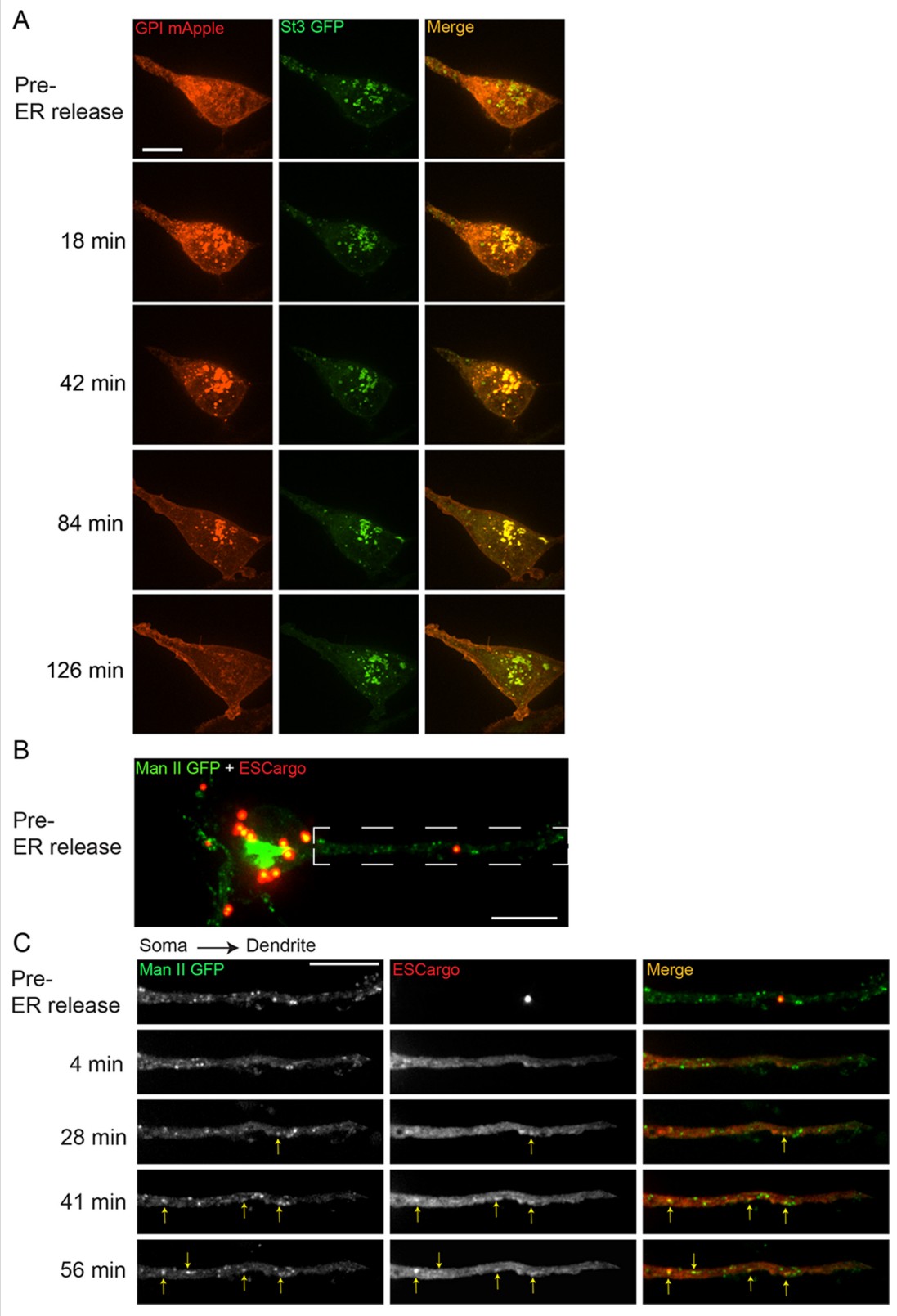

**Figure 3.** Released endoplasmic reticulum (ER) cargo traffic through Golgi satellites to the cell surface in nicotine-treated α4β2R-expressing cells and neurons. (**A**) The retention using selective hooks (RUSH) system was used to synchronize release of ER cargo (RUSH-GPI-mApple) in nicotine-treated (10 mM, 17 hr) α4β2R cells. Cells were transfected with St3-GFP and RUSH-GPI-mApple (retained in the ER via an ER-targeted streptavidin/streptavidin binding peptide hook) for 24 hr and then treated with 10 µM nicotine for 17 hr. Scale bar, 10 µm. GPI-mApple traffics through dispersed Golgi

*Figure 3 continued on next page*

*Figure 3 continued*

elements following biotin-mediated release from ER. (**B–C**) Released ER cargo traffic through Golgi satellites in dendrites. Cortical cultures (DIV 10) were transfected with Man II-GFP, a modified ESCargo that forms aggregates in the ER lumen, and HA-tagged α4β2R subunits. After 24 hr, neurons were treated with nicotine for 17 hr and then imaged for Man II-GFP (green) and ESCargo (red), which is aggregated prior to ER release. Scale bar, 10 μm. (**B**) Image of the soma and dendrite before cargo release from the ER. Scale bar, 10 μm. (**C**) Live imaging of the dendrite boxed in B. Displayed are the ER cargo, ESCargo (red), and the Golgi satellites, marked by Man II-GFP (green) at the indicated times before (pre-release) and after addition of a synthetic ligand that dissolves the ER aggregates, allowing ER exit. Cargo trafficking through Golgi satellites are marked by yellow arrows. Image frames were acquired every 4 min for 1 hr. Scale bar, 10 μm.

The online version of this article includes the following figure supplement(s) for figure 3:

**Figure supplement 1.** α4β2R cells were transfected with St3-GFP and retention using selective hooks (RUSH)-GPI-mApple (retained in the endoplasmic reticulum (ER) via an ER-targeted streptavidin/streptavidin binding peptide hook) for 24 hr.

## Somatic and dendritic Golgi satellites localize near ERESs and endosomes

We next focused on whether dispersed Golgi satellites in dendrites arising under nicotine treatment were localized near ER export sites (ERESs) and recycling endosomes, as reported in nonstimulated neurons (*Mikhaylova et al., 2016*). This Golgi satellite positioning would minimize the extent that transport vesicles would need to travel to transit cargo from ER to plasma membrane. To identify ERESs in relation to Golgi, nicotine-treated α4β2R-expressing neuronal cultures were co-transfected with St3-GFP to label Golgi membranes, and with the COPII component mCh-Sec23 to label ERESs. To identify endosomes in relation to Golgi, α4β2R-expressing neuronal cultures transfected with St3-GFP were stained with an endosome-specific antibody, anti-EEA1.

St3-GFP-labeled Golgi puncta were often found to reside next to mCh-Sec23 structures in dendrites of nicotine-treated neurons (*Figure 4A*, see arrows). A line scan analysis across the boxed structure in *Figure 4A* revealed that the peak intensity of St3-labeled Golgi puncta was ~300 nm from the peak intensity of ERESs marked by Sec23 (*Figure 4B*). Dendritic Golgi satellites in nicotine-treated neurons were also adjacent to early endosomes labeled with the early endosomal marker EEA1 (*Figure 4C*, see arrows), with an individual Golgi satellite and endosome within 300 nm of each other's center (*Figure 4D*). The number of Golgi satellites closely associated with endosomes was increased by neuronal stimulation (see Figure 8E,F). Additional experiments were performed to examine whether Golgi satellites and endosomes are separate organelles. We found little overlap between St3 staining and other endosomal markers (~25%) while much higher overlap was observed between St3 and other Golgi markers (*Figure 4—figure supplements 1 and 2*).

In triple-labeling experiments, in which neurons were transfected with St3-GFP and mCh-Sec23 and then labeled with antibodies to EEA1, we found that many Golgi satellites resided adjacent to both Sec23-labeled and EEA1-labeled puncta (*Figure 4E and F*). Thus, dendritic Golgi satellites often situate themselves next to sites of ER export and early endosomes, similar to what has been reported for dendritic Golgi satellites in untreated neurons (*Mikhaylova et al., 2016*).

## Nicotine-induced Golgi fragmentation resembles Golgi fragmentation after microtubule depolymerization

When microtubules are depolymerized by nocodazole treatment, Golgi enzymes are relocated to ERESs that are still functional for secretory transport (*Cole et al., 1996*). Because we found that Golgi fragmentation induced by nicotine treatment also leads to Golgi enzymes relocating to transport competent structures near ERESs, we tested whether other features of Golgi fragmentation seen during nicotine treatment were similar to those observed during nocodazole treatment. Golgi fragments induced by either nicotine or nocodazole treatment in non-neuronal cells expressing α4β2Rs had similar size ranges and contents, with some fragments containing both St3 and GM130 (usually larger ones) and others having only St3 (often small ones) (*Figure 5A and B*; *Figure 5—figure supplement 1*). In addition, Golgi fragmentation induced by either treatment was reversible upon drug removal (*Figure 1—figure supplement 1*). Despite these similarities, microtubules (visualized using the microtubule binding protein ensconsin) remained intact when either α4β2R-expressing HEK293 cells or neurons were treated with nicotine, unlike that observed when microtubules were depolymerized with nocodazole (*Figure 5C–E*). This suggested that Golgi fragmentation in nicotine-treated

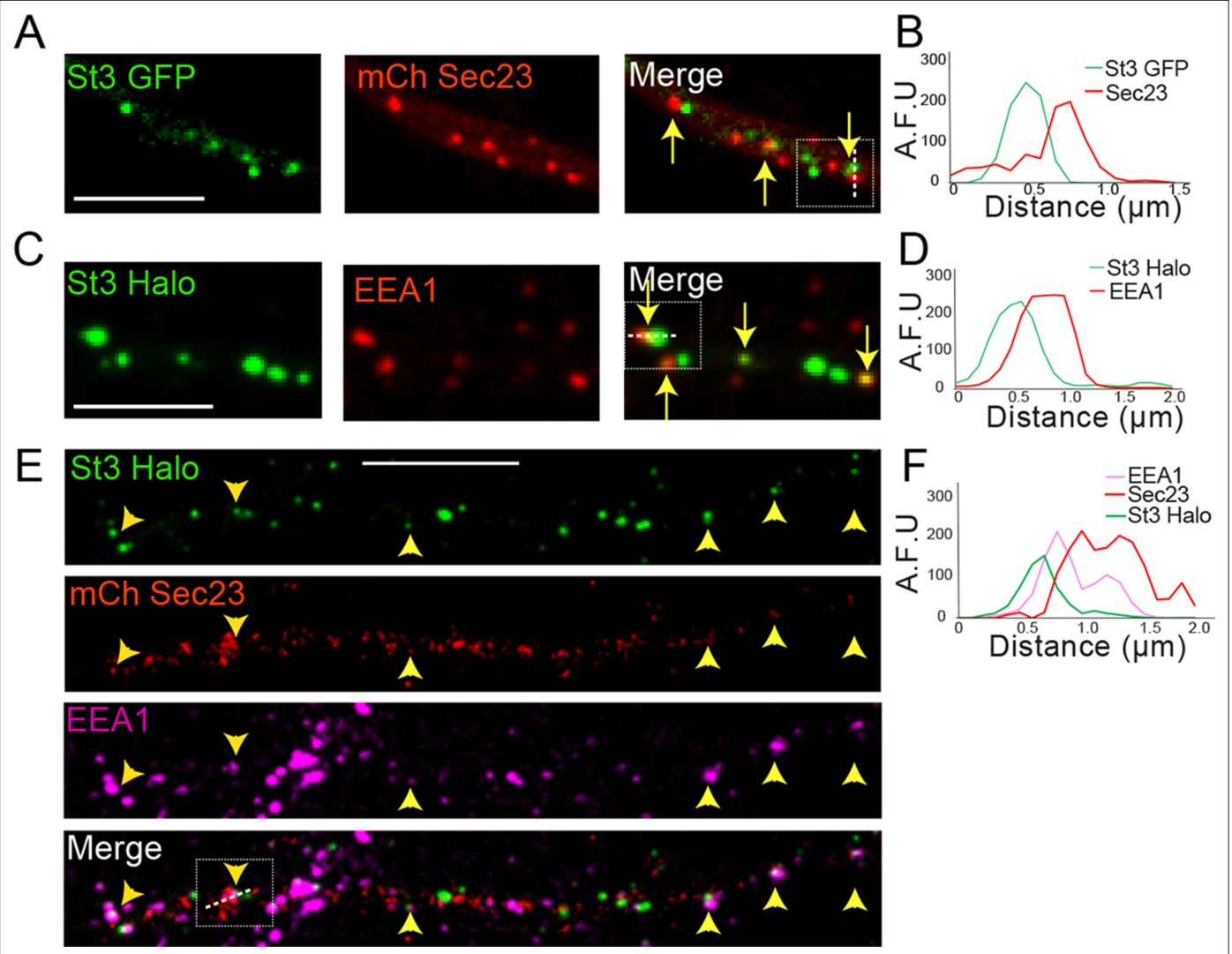

**Figure 4.** Dendritic Golgi satellites localize near endoplasmic reticulum exit sites (ERESs) and endosomes. (**A**) Golgi satellites (St3-GFP) frequently pair with ERESs (mChSec 23) in dendrites. Primary cortical cultures were transfected with St3-GFP, mCh-Sec23, and HA-tagged α4β2R subunits. Neurons were treated with nicotine for 17 hr and fixed. Dendrites were imaged for Golgi satellites (St3-GFP, green) and ERESs (mChSec 23, red). Closely aligned Golgi satellites and ERESs are marked by yellow arrows. Scale bar, 5 μm. (**B**) An example of the close association between Golgi satellites (green) and ERESs (red). Signal intensity of the Golgi satellite and ERESs measured in arbitrary fluorescent units (AFU) in a line scan through the pair in the boxed area. (**C**) Golgi satellites (St3-GFP) frequently pair with early endosomes (EEA1 staining) in dendrites. Neurons were transfected with St3-GFP and after nicotine treatment, fixed and stained with antibodies to EEA1 before imaging as in A. Golgi satellites closely aligned with early endosomes are marked by yellow arrows. Scale bar, 5 μm. (**D**) An example of the close association between Golgi satellites (green) and early endosomes (red). Signal intensity of the two compartments was measured and displayed as in A. (**E**) Golgi satellites (St3-Halo) frequently form triads with ERESs (mChSec 23) and early endosomes (EEA1 staining) in dendrites. Neurons were transfected with St3-GFP, mCh-Sec23, and treated with nicotine for 17 hr before fixation and antibody labeling for EEA1 and imaging. Golgi puncta (green) closely aligned with ERESs (red) and early endosomes (mauve) are marked by yellow arrow heads. Scale bar, 10 μm. (**F**) An example of the close association between Golgi satellites (green), ERESs (red), and early endosomes (mauve). Signal intensity for markers of the three organelles was measured and displayed as in A and C.

The online version of this article includes the following figure supplement(s) for figure 4:

**Figure supplement 1.** Moderate exogenous expression of the Golgi enzymes, Man II-GFP, GalNac-T2-mCherry, and St3-Halo in pairs, or singly, along with labeling for internalized fluorescently tagged wheat germ agglutinin (WGA) or an St3 antibody.

**Figure supplement 2.** Measuring the punctal overlap between endogenous or expressed St3, and the endosomal markers, EEA1, internalized transferrin receptor, or VPS35.

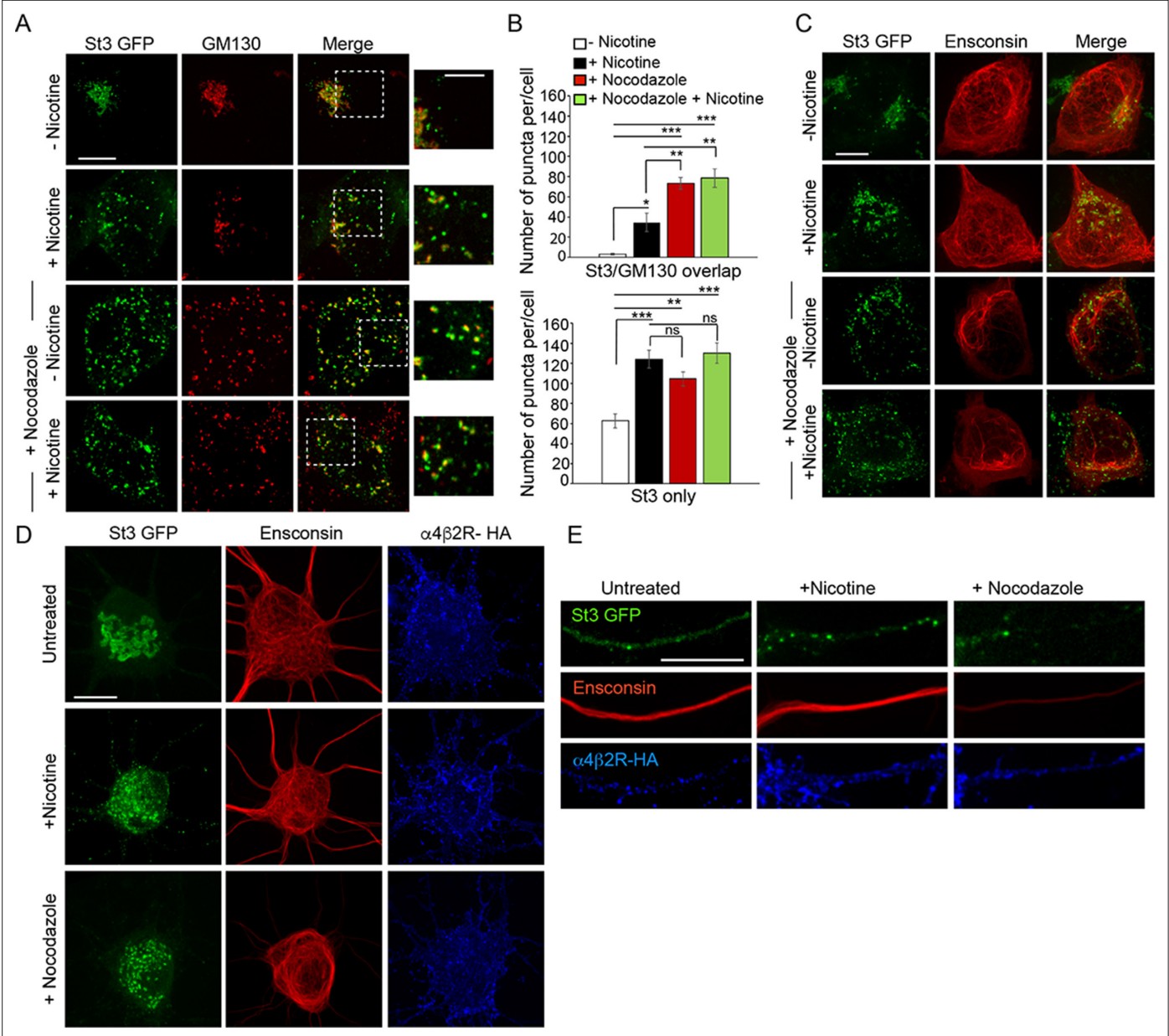

**Figure 5.** Similarities between Golgi dispersal induced by nicotine and nocodazole. (**A**) Comparison of Golgi fragmentation in nocodazole-treated and/or nicotine-treated α4β2R cells. Cells were transfected with St3-GFP (green). Cells were treated (or left untreated) with 10 μM nicotine for 17 hr and further treated for 4 hr with 25 μM nocodazole. Cells were fixed, permeabilized, and immunostained with anti-GM130 antibody (red). Scale bar, 10 μm. Inset scale bar, 5 μm. (**B**) Quantification of the number of puncta per cell displaying St3/GM130 overlap (top) or St3 only (bottom). Data are shown as mean ± SEM, for St3/GM130 overlap, control cells, 3.1 ± 1.0; nicotine cells, 34.6 ± 9.0; nocodazole cells, 73.3 ± 6.0; nicotine and nocodazole cells, 78.5 ± 9.3 (n = 10 cells per group, con vs nic, *p < 0.035; con vs noc, ***p < 0.00003; con vs nic+ noc, ***p < 0.000003; nic vs noc, p < 0.007; nic vs nic+ noc, p < 0.003) and for St3 only, control cells, 62.6 ± 6.9; nicotine cells, 124.1 ± 8.9; nocodazole cells, 104.5 ± 6.9; nicotine and nocodazole cells, 130 ± 10 (n = 10 cells per group, con vs nic, ***p < 0.00005; con vs noc, **p < 0.004; con vs nic+ noc, ***p < 0.000005). (**C**) Effect of nicotine on microtubule stability. α4β2R cells were transfected with St3-GFP (green) and the microtubule binding protein, Ensconsin-mCherry (red). Cells were treated with nicotine and/or nocodazole as in A. Scale bar, 10 μm. (**D–E**) Comparison of the effect of nicotine and nocodazole on primary cortical cultures. Cultures of neurons (DIV 10) were transfected with HA-tagged α4β2R subunits, St3-GFP and Ensconsin-mCherry for 24 hr. Neurons were untreated, treated with 1 μM nicotine or 8 μM nocodazole for 17 hr. Surface α4β2Rs on neurons were labeled with anti-HA antibody (α4β2R-HA; blue). Scale bars, 10 μm. (**D**) Images of Ensconsin-mCherry (red), St3-GFP (green), and α4β2R-HA (blue) in the somata of untreated (top), nicotine- (middle) and nocodazole- (bottom) treated neurons. (**E**) Same as in D except for dendrites.

The online version of this article includes the following figure supplement(s) for figure 5:

**Figure supplement 1.** Histograms displaying the size distribution of St3/GM130 (left) and St3-only (right) puncta (8–10 cells per group).

cells, while sharing similarities to Golgi fragmentation in nocodazole-treated cells, was not due to microtubules being disrupted.

## Glycans on α4β2 receptors become modified to complex forms in nicotine-treated cells

Because Golgi satellites/fragments in nicotine-treated neurons or non-neuronal cells expressing α4β2Rs contained medial/late-acting Golgi enzymes, including Man II, GalNac-T2 and St3, we asked whether these enzymes could modify glycoprotein receptors, in particular α4β2Rs, presumably by passage through Golgi satellites/fragments. To address this possibility, we used gel electrophoresis and immunoblotting to assay molecular weight changes to the α4 subunit of α4β2Rs residing on the plasma membrane of non-neuronal, α4β2R-expressing cells upon treatment with or without nicotine. The surface pool of α4β2Rs was analyzed in these experiments using surface biotinylation followed by streptavidin pull down and immunoblotting with antibodies to the α4 subunit of α4β2R.

As can be seen in *Figure 6A*, after 17 hr of nicotine treatment, the majority of α4 subunits associated with surface-labeled α4β2Rs had shifted to a higher molecular weight form. The differences in α4 subunit band intensities (-Nic vs. +Nic) were likely due to nicotine treatment changing the conformation and/or expression levels of α4β2Rs, as previously shown (*Govind et al., 2012*). The α4 subunits could be cleaved to lower molecular, nonglycosylated forms by the glycan-specific cleavage reagents Endo H or PNGase F (*Figure 6A*). No such shift in molecular weight of surface-labeled α4β2Rs was seen in untreated cells. Incubating cells with swainsonine, a drug that prevents *N*-glycans on proteins from being converted from high mannose to complex forms, prevented the molecular weight shift in α4 subunits during nicotine treatment (*Figure 6B*). Further analysis of the sugar modifications on α4 subunits of α4β2Rs in nicotine-treated cells revealed that they were sensitive to neuraminidase (*Figure 6C*), an enzyme that cleaves off sialic acid from glycans. In addition, α4 subunits could be precipitated by the sialic acid binding lectin SNA (*Figure 6D*), confirming the glycans on α4 subunits contained sialic acid. The increase in the molecular weight of α4 subunits from sialic acid addition was blocked by ammonium chloride treatment, which increases pH within intracellular organelles (*Figure 6E*), raising the possibility that low pH-requiring Golgi sialotransferases St3 and/or St6 were involved in the sialic acid addition (*Rivinoja et al., 2009*).

To test whether the glycan conversion to complex forms correlated with nicotine-induced Golgi fragmentation, we examined conversion of the surface α4 subunits as a function of time during nicotine treatment (*Figure 6F*). Cells expressing α4β2R were surface biotinylated at different times after nicotine treatment and biotinylated α4 subunits were then isolated using streptavidin and analyzed by immunoblotting. Within 2 hr of nicotine treatment, α4 subunits associated with α4β2Rs at the cell surface had begun to display complex glycan forms (*Figure 6F*), with the kinetics of this process (*Figure 6G*) resembling the timing of Golgi dispersal in these cells (see *Figure 1A*). Collectively, these data indicate that *N*-glycans on surface α4β2Rs change from being immature types to mature types containing sialic acid in nicotine-treated cells, with conversion likely occurring as α4β2Rs traffic through dispersed Golgi elements.

## Surface α4β2 receptors are endocytosed and trafficked to Golgi satellites for glycan modification during nicotine treatment

The occurrence of a significant shift in the pool of modified sugars on α4 subunits of α4β2Rs within a few hours of nicotine treatment led us to ask whether activity-induced formation of Golgi satellites/fragments in neurons might be responsible for such glycan remodeling through endocytosis and delivery to Golgi satellites. To explore this possibility, neurons co-expressing HA-tagged α4β2Rs and St3-GFP were allowed to bind and take up fluorescently labeled antibodies to HA in the presence or absence of nicotine. Surface-bound Abs were then removed by acid washing so that only endocytosed α4β2Rs were imaged. After this treatment, significant overlap between endocytosed HA antibody-labeled α4β2Rs and small Golgi puncta in somata and dendrites could be seen, with seemingly greater internalization and Golgi delivery under nicotine treatment (*Figure 7A*). In dendrites, where overlap was easier to quantify, the number of Golgi fragments containing HA antibodies to α4β2Rs increased threefold after nicotine treatment (*Figure 7B and C*). We also found that endocytosed α4β2Rs co-localized with EEA1-labeled endosomes and that this co-localization increased with

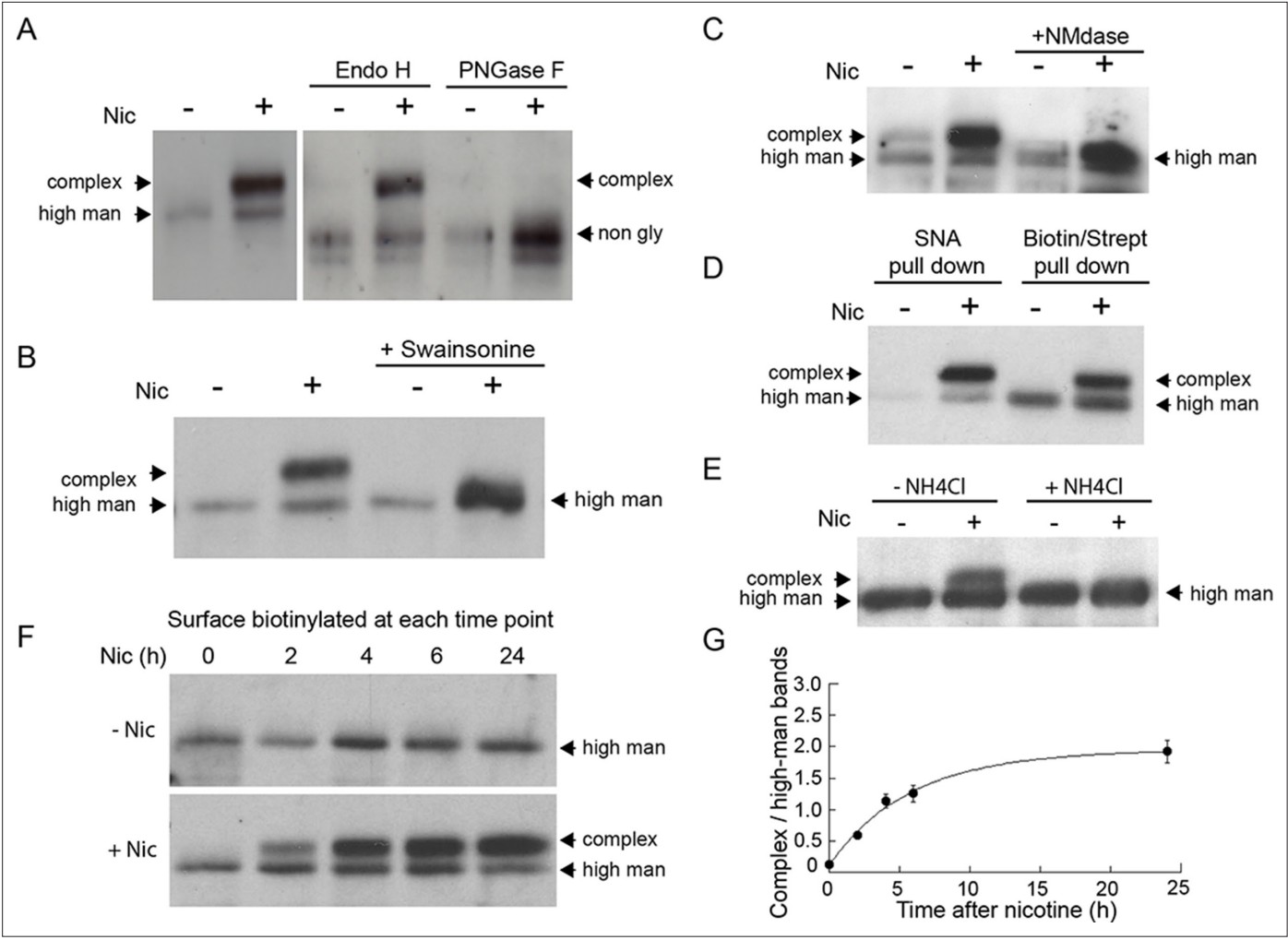

**Figure 6.** Nicotine exposure induces the modification of α4β2R *N*-linked glycans to complex forms. (**A**) Endo H and PNGase F cleavage of surface α4 subunits from untreated or nicotine-treated α4β2R cells. Cells were untreated or nicotine-treated for 17 hr. Proteins on the cell surface were then biotinylated, solubilized, precipitated with streptavidin agarose and glycosidase-treated with Endo H or PNGase F enzymes on the agarose. Afterward, eluted proteins were analyzed on sodium dodecyl sulfate (SDS)-polyacrylamide gel (PAGE) and immunoblotted using anti-α4 antibody. (**B**) Swainsonine treatment blocks nicotine-induced glycan modification of surface α4 subunits. α4β2R cells were treated with the α-mannosidase inhibitor, swainsonine, for 2 hr and then swainsonine and nicotine for 17 hr. Afterward, samples were prepared as in A. (**C**) Neuraminidase (NMdase) cleavage of surface α4 subunits from untreated (-nicotine) or nicotine-treated (+nicotine) α4β2R cells. α4β2R cells were prepared as in A except with NMdase, which cleaves sialic acid, replacing the other glycosidase enzymes. (**D**) Sambucus Nigra (SNA) lectin recognizes α4 subunits after nicotine treatment. Samples were prepared as in A, and after solubilization, proteins were precipitated with the agarose-conjugated sialic acid-recognizing, lectin SNA and then analyzed by immunoblotting with anti-α4 antibody (left) and compared to samples precipitated with with streptavidin agarose as in A–C (right). (**E**) Ammonium chloride treatment blocks nicotine-induced glycan modification of surface α4 subunits. α4β2 cells were treated with or without ammonium chloride (NH$_4$Cl; 10 mM) to inhibit activity of sialyltransferases that require an acidic environment to function. Samples were processed as in B, with NH$_4$Cl replacing swainsonine. (**F**) Time course of surface α4 subunit glycan modification. α4β2R cells were nicotine-treated (+Nic) or untreated (-Nic) for displayed times, after which the cells were surface biotinylated and surface α4 subunits processed as in the previous panels. (**G**) Quantification of the time course of surface α4 subunit glycan modification. Densitometry of α4 subunit bands from three separate experiments, as in F, was preformed and the ratio of complex-trimmed (upper) band to the high-mannose (lower) bands are plotted as the mean ± SEM.

nicotine treatment (*Figure 7—figure supplement 1*). These results demonstrate that endocytosed α4β2Rs can be delivered to Golgi satellites and that nicotine treatment increases this process.

To address whether α4β2Rs that are endocytosed and delivered to Golgi fragments during nicotine treatment undergo processing by Golgi enzymes, we used α4β2R-expressing HEK293 cells to assay for changes in α4 subunit glycosylation. In these experiments, cells were surface biotinylated at time 0 and biotinylated α4 subunits were then pulled down by streptavidin and analyzed by immunoblotting at different time points in the presence or absence of nicotine. We found that within 2 hr after surface

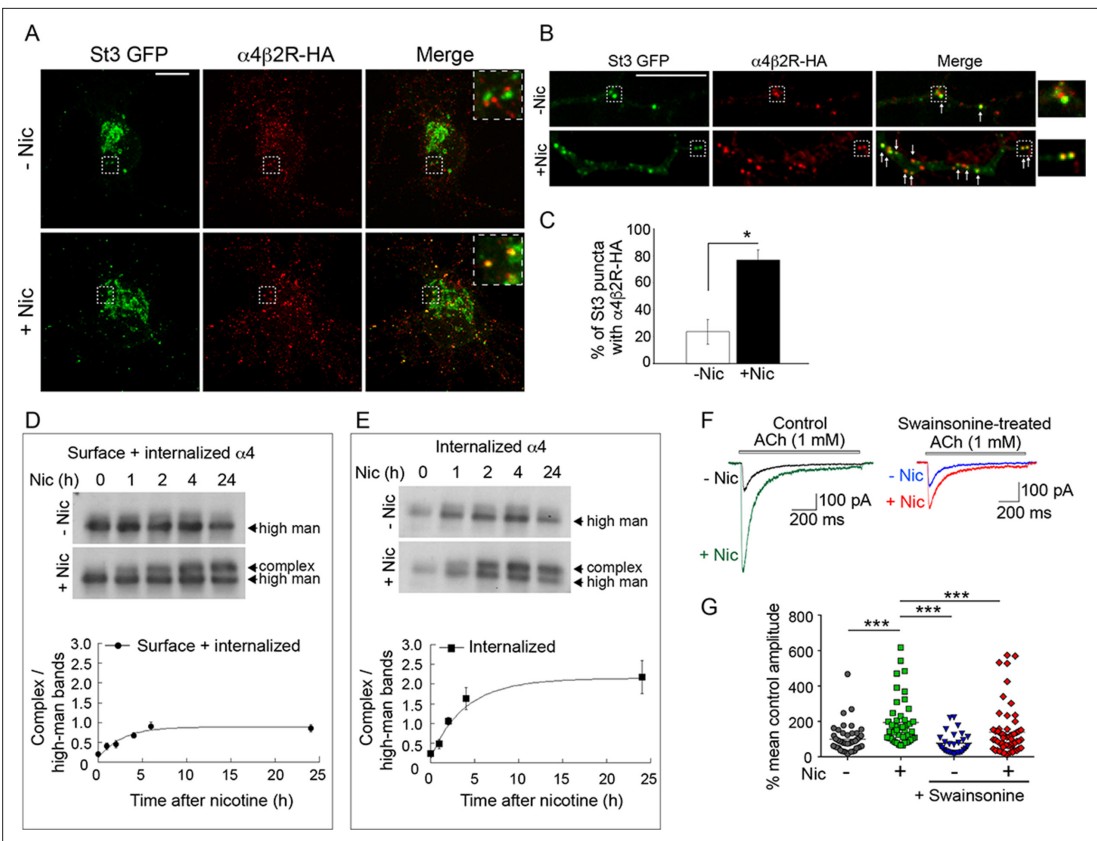

**Figure 7.** Surface α4β2Rs glycan modification and functional changes after endocytosis and trafficking to Golgi satellites. (**A–B**) Endocytosed α4β2Rs co-localize with Golgi satellites in the somata (**A**) and dendrites (**B**) of neurons. Scale bars, 10 μm. Cortical neurons were transfected with St3-GFP (green) and HA-tagged α4β2R subunits and treated with (+Nic) or without (-Nic) 1 μM nicotine for 17 hr. To measure α4β2R endocytosis, cultures were labeled with anti-HA Ab for 30 min, washed, and cells incubated at 37 °C for 2 hr. Afterward, cells were acid washed to remove surface receptor Abs, fixed, permeabilized, and endocytosed α4β2Rs visualized (red). Arrows mark where endocytosed α4β2Rs co-localize with Golgi satellites. (**C**) Quantification of the percentage of Golgi satellites that co-localized with endocytosed α4β2R. Data are displayed as mean ± SEM, control cells, 23.8 ± 9.0; nicotine cells, 76.9 ± 7.5 (n = 7–11 neurons per group, *p < 0.0002). Quantitative analysis was conducted on three independent culture preparations. (**D**) Time course of the α4 subunit glycan modification of the surface and internalized pools of α4β2Rs. Top: α4β2R cells were surface biotinylated with cleavable sulfo-NHS S-S biotin at time 0. Cultures were incubated at 37 °C and followed in the presence or absence of 10 μM nicotine for the indicated times (hr). At each time point, biotinylated proteins, both cell surface and internalized/endocytosed, were precipitated with streptavidin agarose, and immunoblotted with anti-α4 antibody. Bottom: Densitometry of α4 subunit bands from four separate experiments as in top panel and plotted as in *Figure 6G*. (**E**) Time course of the α4 subunit glycan modification of the internalized pools of α4β2Rs. Top: Surface biotinylation was performed as in D at time 0. At each time point, surface biotin was cleaved using glutathione. The remaining internalized/endocytosed biotinylated receptors were isolated using streptavidin agarose and analyzed using immunoblot. Bottom: Densitometry of α4 subunit bands from three separate experiments as in top panel and plotted as in *Figure 6G*. (**F–G**) Block of α4β2R glycan modification by α-mannosidase inhibitor, swainsonine, prevents α4β2R functional upregulation by nicotine. (**F**) Swainsonine treatment blocks nicotine-induced increases in α4β2R current responses. Control (left): A 17–20 hr treatment with nicotine (Nic, 10 μM) induced an approximately fivefold, increase in ACh-evoked (1 mM ACh) current amplitudes in α4β2R-expressing HEK cells. Swainsonine-treated (right): ACh-evoked (1 mM ACh) current amplitudes for swainsonine and nicotine-treated cells. α4β2R cells were treated with swainsonine, for 2 hr and then swainsonine and nicotine for 17 hr. (**G**) Scatter plot of all ACh-evoked current amplitudes plotted as the percentage of mean control current amplitude. Control vs. nicotine p < 0.0001; nicotine vs. control and swainsonine-treated p < 0.001; nicotine vs. nicotine and swainsonine-treated p = 0.0054; control and swainsonine-treated vs. nicotine and swainsonine-treated p = 0.082 (not significant).

The online version of this article includes the following figure supplement(s) for figure 7:

**Figure supplement 1.** α4β2R-HA co-locolizes with EEA1-labeled endosomes in neurons 5 hr after endocytosis.

**Figure supplement 2.** Effect of swainsonine on α4β2R surface expression.

biotinylation and continuing thereafter, the glycans on α4 subunits in nicotine-treated cells had begun being converted to higher molecular weight, complex forms (*Figure 7D*), consistent with their having been delivered to Golgi fragments from the cell surface. No such conversion was seen in untreated cells.

In a different set of experiments, we biotinylated the surface of HEK293 cells expressing α4β2R at time 0 and then measured the pool of biotinylated α4 subunits that was resistant to biotin cleavage (by a membrane impermeable cleavage reagent) over time. As shown in *Figure 7E*, the 'protected' (i.e., internalized) pool of biotinylated α4 (i.e., internalized) in nicotine-treated cells increased over time and by 2 hr consisted of α4 subunits with complex-type glycans of increased molecular weight. No such shift was seen in untreated cells. Therefore, nicotine appears to stimulate surface α4β2Rs to traffic to Golgi satellites in HEK293 cells, where their *N*-glycans are modified to higher molecular weight, complex forms.

## Changes to glycan patterning on α4β2 receptors impact their function

We next explored whether the observed nicotine-induced modifications of *N*-glycans on α4 subunits impacted the overall functioning of α4β2Rs. To address this, we measured nicotine-induced increases in currents in HEK293 cells expressing α4β2Rs and looked to see if they were reduced when swainsonine was added to block glycans on α4β2R from being processed to complex forms. The expressing cells showed significantly enhanced ACh-evoked currents after nicotine treatment (*Figure 7F and G*, control). Notably, this enhancement was markedly reduced by swainsonine treatment without impacting the low level currents under no nicotine treatment (*Figure 7F and G*, swainsonine). Other experiments showed that overall levels of surface α4β2Rs did not change during swainsonine treatment (*Figure 7—figure supplement 2*). These findings thus demonstrate that *N*-glycan repatterning during nicotine treatment, specifically conversion to complex-type forms containing sialic acid, is an underlying factor in facilitating increased α4β2R activity (i.e., functional upregulation) in nicotine-treated, α4β2R-expressing HEK293 cells.

## Effect of neuronal excitation on lectin binding and uptake

The above results suggested that Golgi satellites/fragments induced by neuronal excitability can alter the surface glycoproteome through changes in sugar modifications on glycoproteins. To study the overall extent of these changes, we compared *N*-glycan patterning of glycoproteins on neuronal surfaces before and after increased neuronal activity using two lectins: concanavalin A (Con-A), which recognizes core-glycosylated but not complex-glycosylated glycoproteins, and WGA, which labels glycoproteins with complex-type glycan chains terminated with sialic acid. Prior work using these lectins has revealed that the majority of glycoproteins on the surface of neurons at steady state are significantly core-glycosylated, without complex branched chain elements that include sialic acid (*Hanus et al., 2016*).

Using WGA to examine glycan patterns on neuronal surfaces, we found that neuronal stimulation by either nicotine or bicuculline treatment led to increased WGA labeling (*Figure 8A–C*). In live cell imaging experiments, we further found that surface-bound WGA was rapidly internalized into St3-labeled structures upon nicotine treatment of neurons containing α4β2Rs (*Figure 8D*). Con-A labeling in these same cells, by contrast, did not appear to be significantly endocytosed as indicated by its lack of overlap with St3 structures (*Figure 8D*), although its staining was clustered, consistent with previous observations of Con-A staining of spines (*Hanus et al., 2016*). The St3-labeled structures increased twofold in abundance in neurons activated by bicuculline and were closely juxtapositioned to EEA1-labeled endosomes (*Figure 8E and F*). This efficient endocytosis of WGA in activated neurons could help explain WGA's widespread use as marker of active neuronal circuits (*Broadwell and Balin, 1985*).

Examining dendrites in neurons expressing St3-Halo and HA-tagged α4β2R, in which HA antibodies and WGA were added to the medium, we found that within a short time, WGA and HA antibody-labeled α4β2Rs began appearing in St3-labeled Golgi puncta, with most of the surface pools of WGA and α4β2Rs delivered into the puncta by 4 hr of nicotine treatment (*Figure 8G*). We also examined WGA endocytosis and delivery to Golgi satellites during bicuculline treatment, comparing it to either untreated cells or nicotine-treated cells. Both nicotine and bicuculline treatment led to increased levels of WGA labeling and delivery to St3-labeled Golgi satellites compared to untreated cells (*Figure 8H,I*). Taken together, the above results suggest that the extent to which WGA binds to and is taken up by neurons is increased during neuronal activation, presumably because Golgi satellites are now generating glycoproteins with complex-type glycans, which would change the glycan landscape at the cell surface to one enriched in sialic acid.

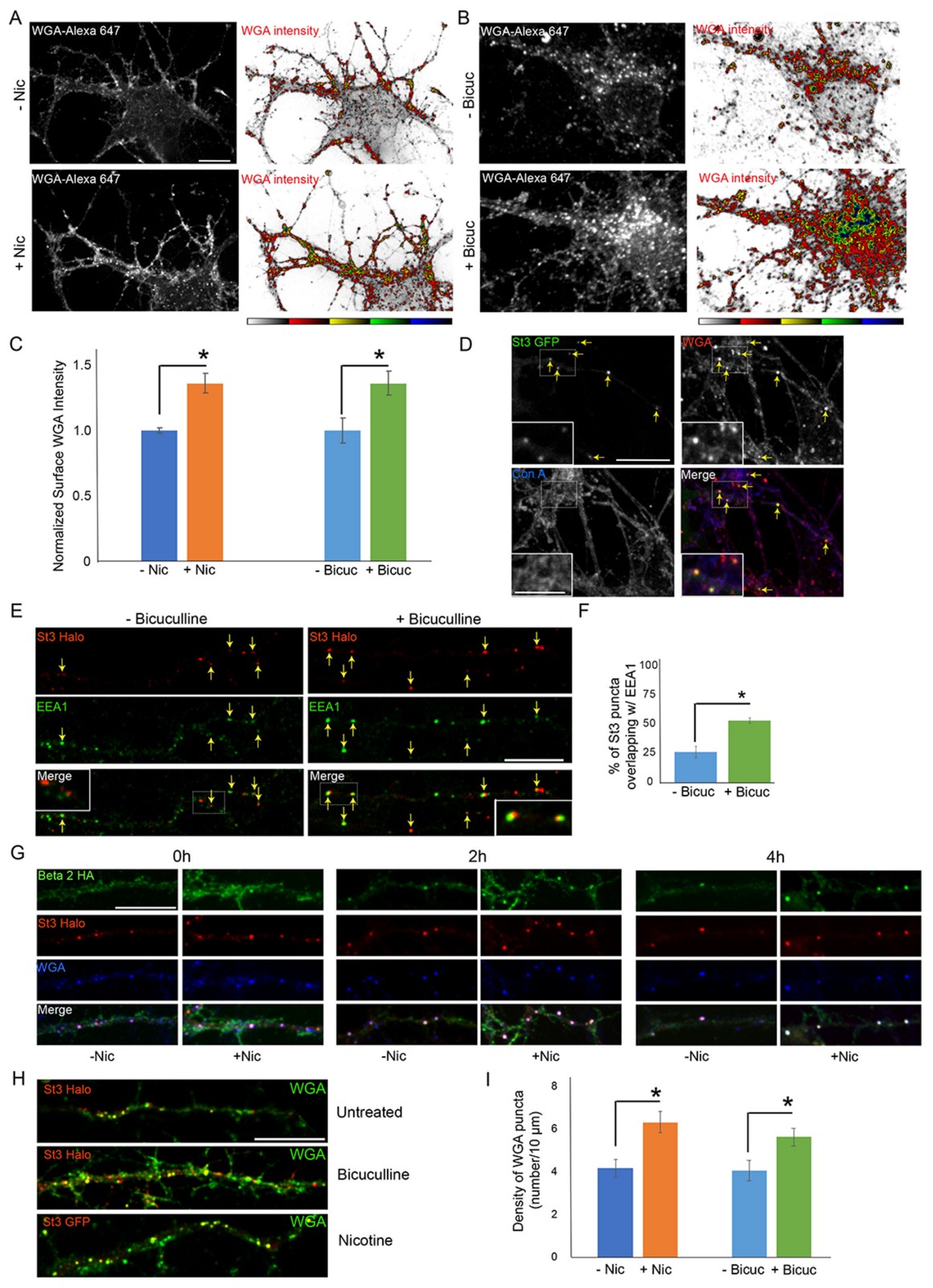

**Figure 8.** Neuronal stimulation increases lectin binding and its endocytosis into Golgi satellites. (**A–B**) Neuronal stimulation by nicotine exposure (**A**) or increased synaptic input with bicuculline treatment (**B**) increased levels of wheat germ agglutinin (WGA) surface staining. Scale bars, 10 µm. (A) Cortical cultures were transfected with HA-tagged α4β2R subunits, treated with or without 1 µM nicotine for 17 hr and cultures were live-labeled with WGA-Alexa 647 for 20 min. Images displaying surface WGA labeling were obtained after cultures were fixed. Pseudocolor intensity profiles are

*Figure 8 continued on next page*

*Figure 8 continued*

shown in the right panels. (**B**) Following overnight bicuculline treatment, untransfected cultures were processed, surface labeled, and displayed as in A. (**C**) Quantification of surface WGA intensities from A and B. Data are shown as normalized mean ± SEM, left; nicotine experiment, control neurons, 1.0 ± 0.02; nicotine-treated neurons, 1.4 ± 0.07 (n = 7–11 neurons per group, *p < 0.0007), right; bicuculline experiment, control neurons, 1.0 ± 0.1; bicuculline-treated neurons, 1.4 ± 0.09 (n = 14–16 neurons per group, *p < 0.006). Quantitative analysis was conducted on two independent culture preparations. (**D**) Differences between Con-A and WGA surface labeling when endocytosis of lectins occurs. Cortical cultures transfected with St3-GFP were surface-labeled with WGA-Alexa 568 and Con A-Alexa 647 for 20 min, washed, then incubated at 37 °C for 2 hr. WGA, but not Con A, was rapidly internalized and overlapped with St3-containing Golgi satellites (yellow arrows). Insets are higher magnification of hatched boxes. Scale bar, 10 µm. Inset scale bar, 2.5 µm. (**E**) Neuronal excitation leads to increased association of dispersed Golgi in dendrites with early endosomal membranes. Cultures were transfected with St3-Halo, treated with or without bicuculline for 17 hr, and fixed, permeabilized, and immunostained with anti-EEA1 antibody (green). Insets are higher magnification of hatched boxes. Scale bar, 10 µm. (**F**) Quantification of the percentage of St3 puncta that overlapped with EEA1. Data are shown as mean ± SEM, control neurons, 26.4 ± 4.8; bicuculline-treated neurons, 53.5 ± 2.3 (n = 10 fields per group, *p < 0.0007). (**G**) WGA is endocytosed with α4β2Rs and traffics into Golgi satellites. Cortical neurons were transfected with St3-Halo, HA-tagged α4β2R subunits and treated with 1 µM nicotine for 17 hr. α4β2R endocytosis was imaged by antibody feeding as in *Figure 7A and B* (anti-HA, green), and ST3-Halo labeled with Halo JF 594 (red), and WGA with WGA-Alexa 647 (blue). After labeling at time 0, neurons were incubated at 37 °C for the indicated times to follow their endocytosis. As in D, WGA showed rapid internalization into St3 vesicles during the 30 min incubation period (0 hr), and over the time course, endocytosed β2HA showed increased co-localization with vesicles containing both WGA and St3-Halo. Scale bar, 10 µm. (**H**) Cultures were transfected with St3-Halo alone, or along with α4 and β2HA subunits (nicotine treatment), for 24 hr, then treated with and without bicuculline or nicotine for 17 hr, and surface labeled for 30 min with Halo JF 594 and WGA-Alexa 647. Cells were washed and incubated further for 2 hr to allow for WGA internalization. Scale bar, 10 µm. (**I**) Quantification of WGA puncta densities from F. Data are shown as mean ± SEM, left; nicotine experiment, control neurons, 4.1 ± 0.4; nicotine-treated neurons, 6.3 ± 0.5 (n = 6 fields per group, *p < 0.00003), right; bicuculline experiment, control neurons, 4.0 ± 0.5; bicuculline-treated neurons, 5.6 ± 0.4 (n = 6–8 fields per group, *p < 0.02). Quantitative analysis was conducted on two independent culture preparations.

## Discussion

Neural plasticity depends on proper biochemical tuning of synaptic receptors, channels, and adhesion molecules on neuronal surfaces to maintain and modulate synaptic function. How this is achieved represents a major question in neuroscience. Our findings reveal that changes in neuronal excitation can reshape the neuronal surface glycoproteome by a process involving the proliferation of Golgi satellites, which we showed function as distal glycosylation stations in dendrites for local glycoprotein processing at the dendritic surface.

In neurons with low electrical activity, we found that there are normally relatively few dendritic Golgi satellites. Golgi enzymes are instead found in tightly clustered Golgi structures in the soma or in a few GM130-containing Golgi outposts localized at dendrite branch points (*Horton and Ehlers, 2003*). As a result, most newly synthesized glycoproteins in dendrites reach the cell surface through a Golgi bypass pathway, involving transfer of glycoproteins from ERESs to plasma membrane via endosomes (*Bowen et al., 2017*), without modification by Golgi enzymes (*Hanus et al., 2016*). The resulting atypical *N*-glycosylation pattern of glycoproteins on the dendritic surface is proposed to be important for controlling membrane protein homeostasis and function in neurons, possibly by accelerating turnover of membrane proteins and modulating synaptic signaling (*Hanus et al., 2016*; *Thayer et al., 2016*). That various features of this system are altered during neuronal excitation has been hinted at from prior work, which showed that during neuronal activity the Golgi apparatus disperses (*Thayer et al., 2013*) and synaptic surface sialic acid content is altered (*Boll et al., 2020*). How these new properties come about, their relationship to each other, and the role(s) they play in excited neurons have remained unclear. In investigating the effects of excitation-induced Golgi fragmentation, we found it led to the formation of dendritic Golgi satellites. Trafficking through these Golgi satellites caused glycans on glycoproteins, like α4β2Rs, to be modified to mature, sialic acid-containing forms, resulting in changes that included the functional upregulation of α4β2Rs.

Evidence supporting this model (depicted in *Figure 9*) came with our characterization of cells expressing nicotinic α4β2Rs, which in response to nicotine treatment become activated and fragment their Golgi. We found that prior to nicotine treatment, α4β2Rs arrived on the plasma membrane in an immature, high-mannose state. With the addition of nicotine, surface α4β2R glycans changed their state, with high-mannose types being converted to complex forms having terminal sialic acid residues. The glycosylation alterations paralleled Golgi fragmentation and Golgi satellite formation in these cells, suggesting glycosylation changes were linked to the formation of Golgi satellites. We further found that swainsonine treatment, which prevents glycans on proteins from being converted to complex forms, blocked nicotine-induced upregulation of α4β2R currents. This suggested that the

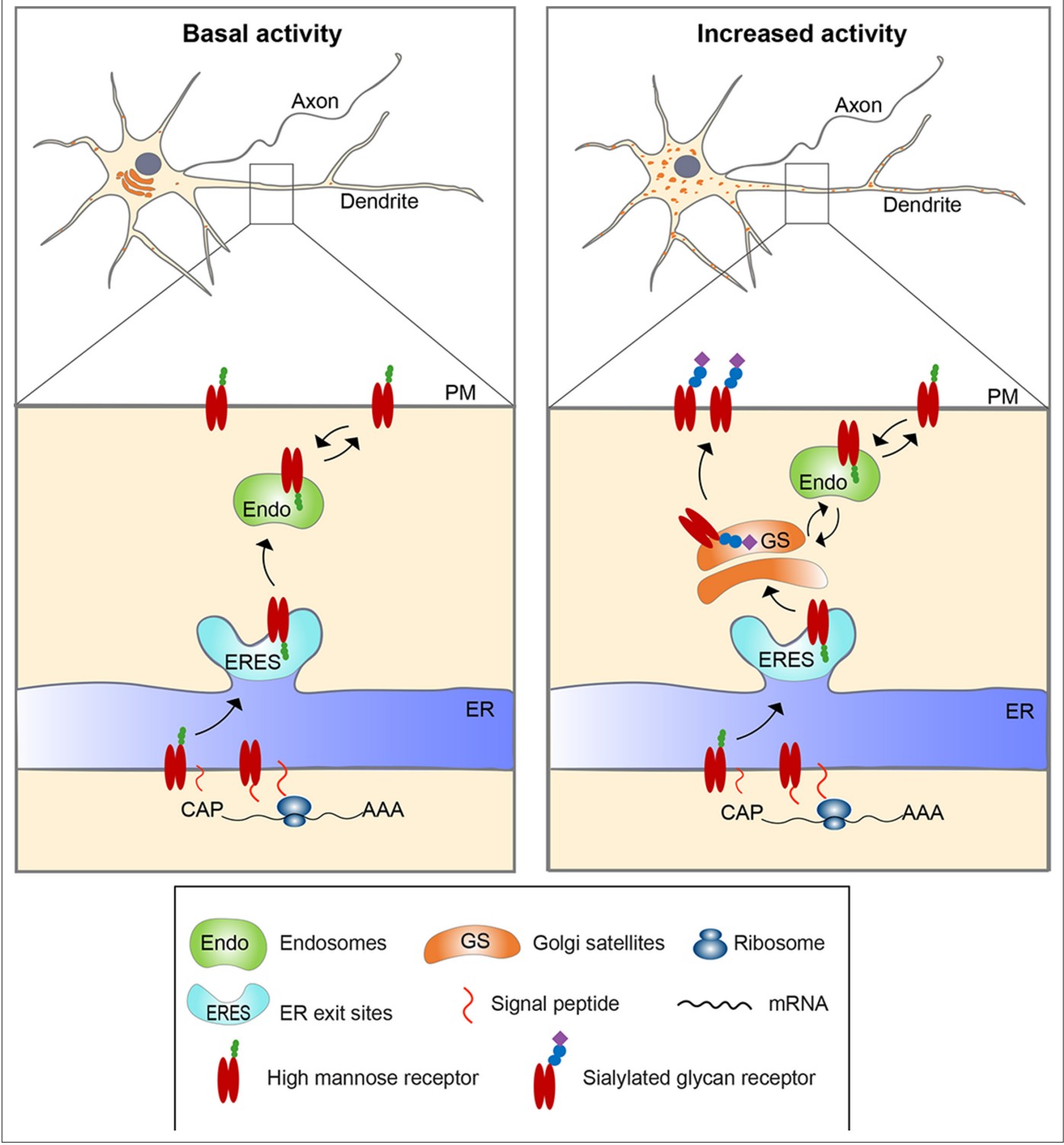

**Figure 9.** Model for activity-induced induction of Golgi satellites in dendrites. In neurons with little excitatory activity, locally synthesized membrane and secreted proteins in dendrites appear to bypass Golgi satellites, trafficking through endosomes to the cell surface, as reported previously (***Bowen et al., 2017***). With increased excitatory activity, increasing numbers of Golgi satellites arise from local endoplasmic reticulum exit sites (ERESs). Locally synthesized membrane and secreted proteins then traffic through these Golgi satellites before reaching the cell surface, having their *N*-linked glycans modified from high-mannose types to sialic acid-containing, complex types. Glycoproteins already on the cell surface can also become modified to complex types by being endocytosed to endosomes that traffic cargo back to the nearby Golgi satellites.

increased α4β2R functionality on the cell surface during nicotine treatment depended on *N*-glycan maturation by Golgi enzymes.

Glycan processing of glycoproteins to sialylated, complex forms in stimulated neurons occurred either during forward trafficking from the Golgi to plasma membrane or after surface delivery by endocytosis and trafficking back to Golgi satellites. The latter was demonstrated using surface biotin-labeled α4β2Rs. When nicotine was added, glycans on surface biotinylated α4β2Rs shifted over time from immature, high-mannose forms to mature, sialylated forms. This shift coincided with α4β2Rs being endocytosed and trafficked back to Golgi satellites as observed in neuronal dendrites. These findings are consistent with a previous study that found that cell surface glycoproteins in Chinese hamster ovary cells can recycle through the trans-Golgi (*Huang and Snider, 1993*).

The extent that different surface glycoproteins become sialylated during neuronal activity remains to be explored. Prior work has shown that kainate receptors in neurons can traffic through Golgi satellites (*Evans et al., 2017*), are sialylated (*Boll et al., 2020*), and undergo functional changes with changes in their glycosylation (*Vernon et al., 2017*). Dynamic regulation of sialylation on synaptic proteins, including various ion channels, ion transporters, receptors, and surface adhesion molecules also has been described (*Scott and Panin, 2014*; *Boll et al., 2020*). Finally, we found a significant increase in binding and uptake of sialic acid-targeted WGA when the lectin was fed to neurons excited by bicuculline or nicotine treatment. Thus, it is possible that a significant proportion of glycoproteins on dendritic surfaces become sialylated when neurons become electrically excited due to Golgi satellite formation.

Why might addition of sialic acid to glycoproteins be significant to stimulated neurons? Sialylated carbohydrate chains are negatively charged and thus can engage in electrostatic interactions with ions and other charged groups on the cell surface, impacting channel and/or receptor activity (*Green and Andersen, 1991*) or affecting other molecular interactions more broadly (*Scott and Panin, 2014*; *Boll et al., 2020*). More intriguingly, sialic acids can act as biological 'masks' (*Varki and Gagneux, 2012*) by preventing a cell from being recognized by macrophages, which often target cells for elimination when terminal galactose residues instead of sialic acid are exposed on their surface glycans. By converting surface glycans to sialylated forms, stimulated neurons might avoid this type of elimination by surrounding, macrophage-like microglial.

Golgi satellite formation during neuronal activity could have significant roles other than boosting the presence of sialic acid on plasma membrane glycans. Their localization near ERESs (which control protein entry into the secretory pathway) and endosomes (which receive proteins from the cell surface) could help bring together local secretory and endosomal components to enhance surface signaling at nearby postsynaptic domains (*Mikhaylova et al., 2016*). In addition, the secretory activity of Golgi satellites could lead to 'tagging' of postsynaptic regions for increased delivery of maturely modified glycoproteins through micro-secretory pathways or endosomal trafficking.

Exactly what signals trigger Golgi fragmentation and initiate Golgi satellite formation in excited neurons remains unclear. Others studies have identified a large number of genes involved in changes in Golgi morphology and glycosylation in response to external signaling (*Chia et al., 2012*). A role of CaM kinase II and/or IV and calcium has been suggested (*Thayer et al., 2013*). Whatever the upstream signal, a conjecture is that the fragmentation process itself results from some effect on microtubule-based Golgi organization and/or trafficking. We found that Golgi fragmentation during nicotine treatment resembled Golgi fragmentation during nocodazole treatment (which disassembles microtubules). Indeed, both types of Golgi fragments are localized adjacent to ERESs and are secretory transport competent. While nicotine treatment did not cause microtubules to depolymerize, it is possible it caused Golgi fragmentation by interfering with the Golgi's ability to act as a microtubule-organizing center (*Wu and Akhmanova, 2017*; *Valenzuela et al., 2020*). CAMSAP2 is a key noncentrosomal microtubule nucleator found on Golgi membranes and is thought to be important in enabling the Golgi to coalesce along microtubules into a single structure (*Yau et al., 2014*; *Jiang et al., 2014*). CAMSAP2 is recruited to Golgi membranes by the pericentrin-like protein AKAP450, which binds to GM130 (*Wu et al., 2016*; *Sanders and Kaverina, 2015*). Interestingly, CAMSAP2's distribution in neurons is disrupted by neuronal activity (*Yau et al., 2014*), and we found that GM130 is absent from newly formed Golgi satellites in excited neurons. We speculate, therefore, that neuronal stimulation interferes with the machinery that drives microtubule organization at Golgi membranes. This would

promote Golgi fragmentation and the formation of Golgi satellites in dendrites, resulting in changes to the neuronal surface glycoproteome.

## Materials and methods

### Antibodies, cDNA constructs, and reagents

Antibodies used were either purchased from commercial suppliers or were generous gifts from various researchers. Antibodies against the following antigens were used: GM130 (monoclonal; BD Transduction Laboratories, San Jose, CA), GM130 (polyclonal; Sigma, St. Louis, MO), HA epitope (HA.11, monoclonal, Biolegend, San Diego, CA), St3Gal3 (polyclonal; Abcam, Boston, MA), EEA1 (monoclonal; BD Transduction Laboratories, San Jose, CA), and α4 subunit-specific polyclonal Ab 6964 (gift from Dr S Rogers, University of Utah, Salt Lake City, UT). The fluorescent secondary antibodies and ligands that were used include Alexa Fluor 488-conjugated goat anti-rabbit IgG, Alexa Fluor 568-conjugated goat anti-mouse IgG, Alexa Fluor 568-conjugated goat anti-rabbit IgG, Alexa Fluor 647-conjugated goat anti-mouse IgG, Alexa Fluor 647-conjugated goat anti-rabbit IgG, Wheat Germ Agglutinin Alexa Fluor 555 and 647, and Concanavilin A Alexa Fluor 647 (Thermo Fisher Scientific, Waltham, MA) and Janelia Fluor 549 Halo ligand (Promega, Madison, WI). Rat α4 and β2 used for generating the stable cell line were provided by Dr Jim Boulter (University of California, Los Angeles, CA). The HA epitope, YPYDVPDYA, and a stop codon were inserted after the last codon of the 3′-translated region of the subunit DNA of the β2 using the extension overlap method as described in *Vallejo et al., 2005*. Other cDNAs generously provided include: Full-length St3-GFP from Dr Sakari Kellokumpu (University of Oulu, Oulu, Finland) (*Rivinoja et al., 2009*), GalNAc-T2-mCherry from Dr Dibbyendu Bhattacharya (Tata Memorial Center, Mumbai, India), GalT-3xGFP (amino acid 1–60, *Cole et al., 1996*), and mCherry-Sec23 from Dr Jennifer Lippincott-Schwartz (HHMI, Janelia Research Campus), GM130-GFP from Dr Christine Suetterlin (University of California, Irvine, CA), pDsRed-ER (Takara Bio, San Jose, CA), RUSH-GPI-mApple, Rush-GPI-Halo, and Ensconsin-mCherry (Addgene, Watertown, MA). For imaging ER bulk flow trafficking to Golgi, a bicistronic plasmid encoding a modified ESCargo (*Casler et al., 2020*) and Man II-GFP was generated in the laboratory of Benjamin Glick (University of Chicago, Chicago, IL). The *Mus musculus* Man IIA1 gene was obtained from BioBasic (Toronto, Canada), PCR-amplified, and inserted into the ESCargo plasmid using in-fusion cloning (Takara Bio, San Jose, CA). St3-Halo was generated by replacing the GFP in St3-GFP with Halo tag (GenScript USA Inc, Piscataway, NJ). The following reagents were used: (-)-Nicotine, Nocodazole, Bicuculline (Sigma-Aldrich, St. Louis, MO), DL-APV (Abcam, Boston, MA), EZ link sulfo NHS SS Biotin and ProLong Gold Antifade Mountant with DAPI (Thermo Fisher Scientific, Waltham, MA).

### Mammalian cell culture and transfection

The human embryonic kidney (HEK293T) cell line stably expressing the large T antigen (tSA201 cells) was from Dr J Kyle (University of Chicago, Chicago, IL) and served as our 'non-neuronal' cell line. This cell line is not in the list of Database of Cross-Contaminated or Misidentified Cell Lines. Using this parent HEK293T cells, a stable cell line expressing rat α4β2 nAChRs was generated in our lab, expressing untagged α4 and C-terminal, HA epitope-tagged β2 subunits (*Vallejo et al., 2005*). Both the parent HEK and stable α4β2R HEK cell lines were maintained in DMEM (Gibco, Life Technologies) with 10 % calf serum (Hyclone, GE Healthcare Life Sciences, Logan, UT) at 37 °C in the presence of 5 % $CO_2$. DMEM was supplemented with Hygromycin (Thermo Fisher Scientific, Waltham, MA) at 0.4 mg/ml for maintaining selection of α4β2R HEK cells. Hoechst staining and immunofluorescent detection were performed periodically to test for mycoplasma contamination. Fresh batches of cells were thawed and maintained only up to 2 months.

HEK cells were grown on 22 mm × 22 mm coverslips (Thermo Fisher Scientific, Waltham, MA) or 35 mm imaging plates (MatTek, Ashland, MA) coated with poly-D-lysine. Stable cells were plated in media without hygromycin for experiments; 75 % confluent cultures were transfected with 0.5 µg DNA of indicated constructs with Lipofectamine 2000 transfection reagent (Thermo Fisher Scientific, Waltham, MA). After 24 hr of transfection, cells were treated with 10 µM nicotine for 17 hr.

## Primary neuronal culture and transfections

Primary cultures of rat cortical neurons were prepared as described (*Govind et al., 2012*) using Neurobasal Media (NBM), 2 % (v/v) B27, and 2 mM L-glutamine (all from Thermo Fisher Scientific, Waltham, MA). Dissociated cortical neurons from E18 Sprague Dawley rat pups were plated on slips or plates coated with poly-D-lysine (Sigma, St. Louis, MO). For live imaging, neurons were plated in 35 mm glass bottom dishes (MatTek, Ashland, MA). Cells were plated at a density of $0.25 \times 10^6$ cells/ml on 35 mm dishes or per well in a six-well plate. Neuronal cultures were transfected at DIV 10 with the Lipofectamine 2000 transfection reagent (Thermo Fisher Scientific, Waltham, MA) according to manufacturer's recommendations. Neurons were transfected with cDNAs of α4, $β2_{HA}$ and various Golgi markers, ER/ERES markers or RUSH constructs; 0.5 µg of each DNA up to a total of 2 µg were used per 35 mm imaging dish or per well of a six-well plate. Twenty-four hours after transfection, neurons were treated with 1 µM nicotine for 17 hr. Bicuculline and DL-APV treatments were performed as described previously (*Thayer et al., 2013*). Briefly, 1 day prior to transfection, neurons were treated with either 200 µM DL-APV or vehicle. After 2 days of treatment, media-containing drug was replaced with conditioned media lacking drug in one group of neurons ('APV withdrawal'), while the second group of neurons that had been treated were left unchanged ('chronic APV'). One day later, all three groups were fixed and stained.

## Immunocytochemistry

Cultured neurons grown on coverslips were fixed with 4 % paraformaldehyde and 4 % sucrose for 10 min, then incubated with blocking solution (2 % glycine, 1 % BSA, 0.2 % gelatin, 0.5 M $NH_4Cl$, PBS) for 1 hr prior to primary and secondary antibody incubations in blocking solution at room temperature (RT). Primary antibody incubations were for 1 hr and secondary antibodies for 45 min unless otherwise mentioned. Surface expressed α4β2 receptors were imaged by live-labeling the cells with mouse anti-HA antibody (1:500) for 40 min prior to fixation. Cells were permeabilized with 0.1 % Triton X-100 followed by fluorescent secondary antibody.

For internalized staining (antibody-feeding assay), primary antibody was added into the culture medium and incubated at 37 °C for 40 min, then cells were washed twice with acid wash buffer (0.5 M NaCl, 0.5 % acetic acid, pH 2) for 15 s each, and fixed. Cells were then incubated with blocking solution and secondary antibody. For permeabilized staining, after fixation, the cells were permeabilized in 0.1 % Triton/PBS for 5–10 min, blocked and stained with primary and secondary antibodies. The stained coverslips were mounted to glass slides with ProLong Gold (Life Technologies) mounting media and left in dark to harden overnight.

Fluorescence images were acquired using a Leica SP5 Tandem Scanner Spectral 2-Photon scanning confocal microscope (Leica Microsystems), or Marianas Yokogawa type spinning disk confocal microscope with back-thinned EMCCD camera. Images were processed and analyzed using ImageJ/Fiji (US National Institutes of Health).

## Biotinylation of cell surface proteins and isolation of the internalized receptors

α4β2R cells were grown on 6 cm plates. Cells were treated with or without 10 µM nicotine for 17 hr and washed thrice with PBS (containing 1 mM $MgCl_2$ and 1 mM $CaCl_2$), and incubated with 0.5 mg/mL Sulfo-NHS-SS-Biotin at RT for 30 min. Cells were washed thrice with 10 mM Tris HCl, pH 7.4 containing 1 mM $CaCl_2$, and 1 mM $MgCl_2$. Cells were lysed immediately to isolate biotinylated surface proteins/receptors using lysis buffer (150 mm NaCl, 5 mm EDTA, 50 mm Tris, pH 7.4, 0.02 % NaN3, plus 1 % Triton X-100) containing protease inhibitors (2 mm phenylmethylsulfonyl fluoride, 2 mm *N*-ethylmaleimide, and chymostatin, pepstatin, leupeptin, and tosyllysine chloromethyl ketone at 10 µg/ml). To isolate internalized biotinylated proteins, cells were maintained in DMEM at 37 °C following surface biotinylation for the indicated time intervals. At each time points, Sulfo-NHS-SS-Biotin on the cell surface was cleaved with glutathione. Cells were then harvested and lysed; biotinylated proteins were pulled down using streptavidin agarose beads (MilliporeSigma, Burlington, MA) and were analyzed on Western blot.

## Glycoprotein analysis

Deglycosylation experiments were performed on biotinylated surface receptors expressed in α4β2R cells. Surface biotinylated proteins immobilized on streptavidin agarose beads were denatured in Denaturing Buffer (New England Biolabs) at 100 °C for 10 min in accordance with the manufacturer's protocol. For peptide-*N*-glycosidase F (PNGase F) treatment, 10× G7 reaction buffer, 10 % (v/v) Nonidet P-40, and 500 units of PNGase F were added. For endoglycosidase H (Endo H) treatment, 10× G5 reaction buffer and 1000 units of Endo H were added. For Neuraminidase treatment 10 × GlycoBuffer and 50 units of Neuraminidase enzyme was added. The samples were incubated at 37 °C either for 2 hr, then heated for 3 min at 80 °C in 2× sodium dodecyl sulfate (SDS) loading buffer.

For the SNA lectin pull down experiments, lysates from α4β2R cells were incubated with agarose-linked Sambucus Nigra lectin, SNA (100 µl; Vector Laboratories) with rotation at 4 °C overnight. The samples were washed three times with lysis buffer containing 0.1 % (w/v) Triton X-100 in Tris-buffered saline (pH 7.4), and then heated for 5 min at 100 °C in 2× SDS loading buffer (75 µl). The proteins were separated by electrophoresis in a 7.5 % SDS-polyacrylamide gel (SDS-PAGE), transferred to a poly-vinylidene difluoride membrane, then incubated with the primary (polyclonal anti-α4 antibody) and HRP-conjugated anti-rabbit secondary antibodies. The signal was detected using enhanced chemiluminescence with Amersham hyperfilm (Sigma, St. Louis, MO). Inhibition of α-Man II was carried out by treating α4β2R cells with 4 µg/ml swainsonine for 17 hr in the presence or absence of 10 µM nicotine. Inhibition of the activity of all Golgi associated glycosyl transferases was carried out by raising the intracellular pH. Cells were incubated with 10 mM ammonium chloride ($NH_4Cl$). Surface-expressed α4 subunits were analyzed by biotinylation/streptavidin agarose pull down and Western blot analysis.

## Image analysis

St3 punctal densities were quantified using maximum z projections of background-subtracted images. Thresholded punctal size and density was analyzed using the Analyze Particle function in Fiji/ImageJ. Analysis of the co-localization or overlap between two fluorescent punctal signals was carried out by background subtracting and thresholding image fields so that only puncta that were twofold greater than background were selected. Co-localizing puncta were evaluated using the Analyze Particle function in Fiji/ImageJ. Results are expressed as mean ± SEM of n samples unless stated otherwise. Experiments were conducted from a minimum of two independent culture preparations, with 5–10 neurons per experimental group. Statistical comparisons were made using two-tailed Student's t-tests or ANOVA/Tukey's post hoc analysis as indicated. Statistical graphs were generated with StatPlus software.

## Electrophysiology

HEK cells stably expressing α4β2 receptors maintained as described above were plated at low density on glass coverslips before treating for 15–18 hr with media-containing nicotine or other drugs. Cells were voltage-clamped in whole-cell configuration at a holding potential of −70 mV using an Axopatch 200B amplifier running pClamp 10 (Molecular Devices; Sunnyvale, CA). Currents were elicited from cells lifted from the coverslip by the fast application of 1 mM ACh using a piezo-ceramic bimorph system with a solution exchange time of ~1 ms. External solution consisted of (in mM): 150 NaCl, 2.8 KCl, 1.8 $CaCl_2$, 1.0 $MgCl_2$, 10 glucose, and 10 HEPES, adjusted to pH 7.3. Internal pipette solution was (in mM): 110 CsF, 30 CsCl, 4 NaCl, 0.5 $CaCl_2$, 10 HEPES, and 5 EGTA, adjusted to pH 7.3. Peak amplitudes and time courses of desensitization were determined by post hoc analysis using Clampfit 10. Statistical differences were tested for using one-way ANOVAs with Tukey's multiple comparison test in GraphPad (La Jolla, CA).

## Acknowledgements

This work was financially supported by NIH RO1 DA035430, DA044760, and DA043361 (WNG) R01 GM104010 (BSG), T32 GM007183 (FV), and Peter F McManus Foundation (WNG). We thank Vytas Bindokas (University of Chicago) and Louie Kerr (Marine Biological Laboratory), Abhishek Kumar, Panagiotis Chandris, and Hari Shroff (NIH/NIBIB) for technical support on microscopy; Luke Lavis (Janelia research campus) for Halo dyes and Dibbyendu Bhattacharya (Tata Memorial Center), Sakari Kellokumpu (University of Oulu), Jim Boulter (University of California), Steven Standley (Western University

of Health Sciences) for cDNA constructs used in this study. We thank U of C student, Briana Turner for assistance. The authors declare that they have no competing interests. We especially thank Dr Karl Matlin for his thorough reading of the manuscript and many insightful comments and suggestions.

## Additional information

### Funding

| Funder | Grant reference number | Author |
|---|---|---|
| National Institutes of Health | DA035430 | William N Green |
| National Institutes of Health | DA044760 | William N Green |
| National Institutes of Health | DA043361 | William N Green |
| National Institutes of Health | GM104010 | Benjamin S Glick |
| National Institutes of Health | GM007183 | Fernando M Valbuena |
| Peter F McManus Foundation | | William N Green |
| Howard Hughes Medical Institute | | Jennifer Lippincott-Schwartz |
| National Institutes of Health | DA 043469 | Theron A Russell |

The funders had no role in study design, data collection and interpretation, or the decision to submit the work for publication.

### Author contributions

Anitha P Govind, Okunola Jeyifous, Conceptualization, Formal analysis, Investigation, Methodology, Visualization, Writing – original draft, Writing – review and editing; Theron A Russell, Formal analysis, Investigation, Visualization, Writing – original draft, Writing – review and editing; Zola Yi, Investigation, Visualization; Aubrey V Weigel, Investigation, Resources; Abhijit Ramaprasad, Investigation; Luke Newell, William Ramos, Formal analysis, Software; Fernando M Valbuena, Jason C Casler, Benjamin S Glick, Methodology, Resources; Jing-Zhi Yan, Formal analysis, Investigation, Visualization; Geoffrey T Swanson, Formal analysis, Supervision, Visualization; Jennifer Lippincott-Schwartz, Conceptualization, Project administration, Resources, Supervision, Writing – original draft, Writing – review and editing; William N Green, Conceptualization, Funding acquisition, Project administration, Supervision, Writing – original draft, Writing – review and editing

### Author ORCIDs

Anitha P Govind (iD) http://orcid.org/0000-0002-5890-2395
Okunola Jeyifous (iD) http://orcid.org/0000-0002-4176-4694
Aubrey V Weigel (iD) http://orcid.org/0000-0003-1694-4420
Jason C Casler (iD) http://orcid.org/0000-0001-9742-9978
Benjamin S Glick (iD) http://orcid.org/0000-0002-7921-1374
Jennifer Lippincott-Schwartz (iD) http://orcid.org/0000-0002-8601-3501
William N Green (iD) http://orcid.org/0000-0003-2167-1391

### Ethics

All animal procedures were approved by the University of Chicago Institutional Animal Care and Use Committee (protocol #72016) and are in accordance with the recommendations of the Panel on Euthanasia of the American Veterinary Medical Association. Strict adherence to AVMA guidelines was followed to prevent pain and suffering of animals.

Decision letter and Author response
Decision letter https://doi.org/10.7554/eLife.68910.sa1
Author response https://doi.org/10.7554/eLife.68910.sa2

## Additional files

### Supplementary files
• Transparent reporting form

### Data availability

Source data files for all quantitative data presented in the current study have been deposited at Dryad. These contain raw data values, statistical summaries, and raw gels for panels in Figures 1, 2, 5, 6, 7, 8, and Figure 1—figure supplement 2, Figure 2—figure supplement 2, Figure 4—figure supplements 1 and 2, and Figure 7—figure supplement 2. The files can be accessed via Dryad (http://doi.org/10.5061/dryad.qjq2bvqg3).

The following dataset was generated:

| Author(s) | Year | Dataset title | Dataset URL | Database and Identifier |
|---|---|---|---|---|
| Jeyifous OJ, Govind AP, Russell TA, Swanson GT, Green WN | 2021 | Govind Jeyifous et al eLife manuscript source data files | http://dx.doi.org/10.5061/dryad.qjq2bvqg3 | Dryad Digital Repository, 10.5061/dryad.qjq2bvqg3 |

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
