## [Decision Letter]

**Acceptance summary:**

Until recently, the organisation of the Golgi apparatus in cells was supposed to be rather stereotypical. Variations in glycosylation were presumed to depend on levels of glycosylation enzymes expression. This study, based on elegant microscopy shows that re-organisation of the secretory pathway controls the glycosylation of surface proteins in dendrites of neurons. In inactive neurites, proteins traffic from the ER to endosomes than cell surface, decorated with unprocessed, high-mannose N-glycans. Upon neuronal stimulation, Golgi satellites form de novo in neurites and carry the elongation and sialylation of glycans. These results provide a mechanistic basis for the activity-dependent changes in neuronal glycoproteome and open the door to further research on how these changes alter the activity or stability of neurites. More generally, they provide further evidence of the functional plasticity of the Golgi apparatus and role in converting extracellular signals into a glycosylation-based response.

**Decision letter after peer review:**

Thank you for submitting your article "Activity-dependent Golgi satellite formation in dendrites reshapes the neuronal surface glycoproteome" for consideration by *eLife*. Your article has been reviewed by 3 peer reviewers, one of whom is a member of our Board of Reviewing Editors, and the evaluation has been overseen by Vivek Malhotra as the Senior Editor. The following individuals involved in review of your submission have agreed to reveal their identity: Fengwei Yu (Reviewer #2); Cyril Hanus (Reviewer #3).

Essential revisions:

1) Please review the reviewer's comments and provide detailed answers.

2) Please address experimentally if possible the concern of using overexpressed proteins to demonstrate the formation of Golgi outposts. If limited by antibodies, please amend the text to recognize this caveat.

2) The claim that changes in glycosylation pattern impact the function of the nicotinic receptor is quite important in this study but not extensively documented. It is thus important that the authors present evidence that the change in activity is occurring in neurons and dependent on altered glycosylation. An approach using knock-down of sialyl-transferases in neurons would be ideal in this case. In general, additional data to support this claim is critical to ground the functional importance of Golgi re-organization.

*Reviewer #1:*

The authors asked how the change in neuronal activity can result in a change in protein glycosylation and what are the functional implications.

They show that organellar remodeling is induced by nicotinic stimulation in both non-neuronal and neuronal cells, with the fragmentation of the Golgi apparatus. In neurons, the fragmentation results in the formation of Golgi satellites in dendrites, populated by glycosylation enzymes. In unstimulated neurons, cell surface proteins are produced in the endoplasmic reticulum of neurites and transported to the cell surface with unmodified, high mannose N-glycans. By contrast, the formation of Golgi satellites in dendrites results in proteins carrying complex N-glycans. These complex N-glycans on the nicotinic receptors correlate with a higher activity/response to stimulation.

It remains unclear how the vesicles containing glycosylation enzymes are forming Golgi satellites where there was none previously. It is not clear whether the main brake to Golgi satellite formation in unstimulated neurons is the centripetal trafficking of Golgi membranes or if other processes are involved.

This is an interesting study that links multiple processes and proposes an integrated model for how neurons mature in response to stimulation by re-organizing spatially the secretory pathway and glycosylation machinery. It is likely to ignite further work in the domain of neuronal plasticity.

The mechanistic insights driving Golgi fragmentation are very limited. The authors are using the term "electrically excited neurons". Is the fragmentation the result of an influx of Ca^2+^ or another process? Is it possible to have some insights from the Hek293 system?

Since the study is about the formation of Golgi outposts and their population with glycosylation enzymes, I think it would be important to specify the sequences of the reporters used. This is important from a membrane trafficking point of view, especially as the authors show that these outposts are not equivalent to full Golgi (no GM130).

For instance, usually, GAL-T reporter is not a full enzyme cloned with a fluorescent protein.

Also important: there seems to be a mistake in the nature (or naming) of GalNac-T2. This name usually applies to the polypeptide N-acetylGalactosamyl transferase, one of the enzymes that initiate mucin-type O-glycosylation. It does not act on N-glycans and is not a "late-acting enzyme" as mentioned in the text. The N-acetyl-glucosaminyl transferases are the ones acting on high Mannose and usually named GNT-I, II, etc…

In addition, the authors should cite other works studying Golgi plasticity and glycosylation in response to external stimuli. For instance, PMCID: PMC3964318: RNAi screening reveals a large signaling network controlling the Golgi apparatus in human cells

The authors do not discuss how the cycle stimulation-increase in terminal glycosylation-higher nicotinic receptor activity might result in an amplification loop stabilizing further Golgi satellites. Conversely, are Golgi satellites reversible?

*Reviewer #2:*

In this manuscript, Govind and colleagues studied glyco-modification of membrane receptors after neuronal activation. They discovered that Golgi undergoes fragmentation and forms satellite structures in the dendrites after neuronal activation, which act as a micro-secretory system to regulate protein glycosylation. They focused on the nicotinic acetylcholine receptors and showed that the N-glycans on these receptors transformed from an immature, highly mannose form to a mature and complex form containing sialic acid upon neuronal activation. The experiments are very well designed and images are convincing. The conclusions are well supported by their data. It is a pleasure to read this beautiful manuscript. Overall, this is an important and innovative study.

There are a few technical issues for the authors to address before publication.

1. GalT, St3, Man II and GalNAc-T2 are all late stage glycosylation enzymes. In Figure 1C, many Golgi satellites are St3 positive and GalNAc-T2 negative in HEK cells. However, in Figure 2B, St3 and GalNAc-T2 fully co-localized in the dendrites of neurons. The authors should explain this discrepancy? Is it due to different cell types or distinct mechanisms? The authors should discuss the possibilities. Do other enzymes such as ManII, also co-localize with St3 in HEK cells?

2. The conclusion that complex glycosylation increased a4b2R receptor functionality should be interpreted cautiously, as shown in Figure 7F. However, In Figure 6B, total protein level of a4b2R is apparently reduced after swainsonine treatment. The authors did not provide any evidence to exclude that reduced a4b2R protein caused lower current amplitude in Figure 7F. I suggest that the authors should discuss this caveat. Alternatively, is it possible to knock down some glycosylation enzymes without affecting the a4b2R protein levels and then check if a4b2R receptor functionality is impaired? However, this experiment is not essential for publication. If the authors have this data, it is nice to include.

3. In Figure 5D-E, St3-GFP seems not to co-localize with α4β2R-HA after nicotin treatment. Are the images representative? Moreover, it is interesting to show if upon nocodazole treatment, St3-GFP and α4β2R-HA colocalized, as shown in the case of nicotin treatment (Figure 7A-B). It is also interesting to test whether α4β2R undergo glyco-modification after nocodazole treatment. These experiments would fully address similarity and difference between nicotin treatment and nocodazole treatment. If the authors already have this data, it is nice to include. Otherwise, the authors can publish them in another future study.

4. In Figure 7D-E, Biochemical data show that the surface receptors undergo endosomal recycling and glyco-remodelling within 5 hours after nicotin treatment. Can the author also show that in cultured HEK cells or neurons, EEA1 colocalizes with α4β2R-HA as early as 5 h after nicotin treatment?

5. It was reported that golgi outposts rarely distributed in axons of fly sensory neurons. Do Golgi satellites localize in axons?

*Reviewer #3:*

In this study, Govind et al., address whether remodeling of neuronal secretory organelles by neuronal activity regulates the trafficking and glycosylation of the nicotinic receptor.

Using HEK cells and cortical neurons stably or transiently overexpressing nicotinic receptor subunits, the authors show that long exposure to nicotine or bicuculline induces a fragmentation of the Golgi apparatus (GA), resulting in increased numbers of so-called Golgi satellites that contain trans-Golgi glycosylation enzymes in dendrites, in particular sialyl transferase 3 (St3), but which are mostly not apposed to endogenous cis-Golgi proteins such as GM130.

Using HEK cells and neurons transfected to express cargo whose exit from the ER can be experimentally controlled by specific ligands, they provide evidence these satellite structures (here labeled with ManII-GFP) engage in forward membrane trafficking and are thus likely functional, hence extending previous observations made by Horton and Ehlers with VSVGts045 and GalT-GFP (Horton et al., 2003).

As shown by co-labelling of endosome markers and internalized nicotinic receptors and lectins, they provide evidence that St3 containing structures are closely apposed to some dendritic endosomes and engage in post-endocytic trafficking, a process regulated by nicotine and neuronal activity.

Finally, using HEK cells, they provide compelling evidence that the fragmentation of the GA is correlated with changes in the glycosylation status of the nicotinic receptor and enhances the production of surface receptors with hybrid/complex N-glycans, hence increasing the amplitudes of nicotine induced currents.

Strengths. This study addresses an important question and describes interesting and exciting findings.

Weaknesses. in the current form of the manuscript, some claims are not fully supported by experimental evidence and would benefit from additional controls. The manuscript could be improved by addressing the following concerns:

1. Potential overexpression artifacts. A lot of confusion in the field has resulted from the use of over-expressed membrane cargo proteins or organelle markers. As seen with endogenous Golgi proteins including GM130, GRASP55/65, giantin … and GFP-tagged markers such as GalT-GFP or ManII-GFP (after moderate expression), Golgi membranes (i.e Golgi outposts or GO) are rarely found in dendrites (see Reviews). Consistent with this, extensive 3D EM reconstruction of dendrites in hippocampal and cortical pyramidal neurons (e.g., Wu et al., 2017, doi: 10.1073/pnas.1701078114, Cui-Wang et al., doi: 10.1016/j.cell.2011.11.056) have shown a high abundance of dendritic ER, ERGIC and endosomes but a very rare occurrence of stacked membranes characteristic of bona fide Golgi elements in dendrites. As shown for example by Horton and Ehlers with GalT-GFP, co-expression of membrane cargo including VSVG greatly enhances the number of GO in dendrites. In a comparable manner, the data shown here in Figure 2C ii versus 3B-C also indicate that increasing secretory load in the secretory pathway induces the formation/recruitment of ManII-GFP containing elements in dendrites. This issue could be addressed by performing additional control experiments.

2. Are Golgi satellites actually Golgi or neuron specific endosomal compartments? The distribution pattern of endogenous St3 shown in Figure 2D strikingly differ from that of other med/late Golgi markers used in the study. In particular, it seems that St3 containing structures (i.e here the endogenous protein) are way more abundant in dendrites than cis/late Golgi membranes labelled with GalT, ManII and GalNacT2. Is this the case? If so, how can the authors rule out that, in neurons, St3 is associated both to the late-Golgi and to non-Golgi membranes such as TGN/endosomal structures or other endomembrane systems, depending on neuronal activity ? Indeed, the number and morphology of St3 containing structures in dendrites is more reminiscent of the endosomal pathway and somehow resemble that of pGolT (Mikhaylova et al., 2016), a synthetic membrane protein containing both TGN/endosomal and Golgi targeting motifs. Rephrasing some statements and performing additional experiments would help addressing this.

3. Functional relevance of data in HEK cells. Deglycosylation assays shown in Figure 6 and 7 are compelling but, although I may have missed something, were performed in HEK cells only. The notion that neuronal activity reshapes the neuronal surface glycoproteome is thus based primarily on increased levels of surface and internalized WGA binding proteins in neurons after treatment with nicotine or bicuculline. The title of the paper is thus somehow misleading. Experiments on the surface expression, glycosylation and function of endogenous nicotinic receptors in neurons would significantly increase the impact of the study.

Major concerns

1. Potential overexpression artifacts. I strongly encourage the authors to address the issue by carefully comparing and quantifying the number of dendritic structures containing endogenous Golgi proteins and/or moderately expressed ManII-GFP or GalT-GFP (see also Minor comment 2) with and without co-expression of nicotinic receptor subunits, St3-GFP or EScargo.

2. Are Golgi satellites actually Golgi or neuron specific endosomal compartments? The authors may want to address this point by assessing the distribution of endogenous and overexpressed St3 in relation to a broader set of endosomal and retromer markers than EEA1 (e.g., internalized transferrin, Vps35, Rabs, …).

3. Functional relevance of data in HEK cells. The claims of the study would be significantly strengthened by addressing whether nicotine or bicuculine regulates the surface expression and glycosylation status of endogenous nicotinic receptors in neurons. Along the same line, including functional data directly addressing whether reported changes in glycosylation impacts neuronal excitability in this context would also enhance the impact of the results.

---

## [Author Response]

Reviewer #1:The authors asked how the change in neuronal activity can result in a change in protein glycosylation and what are the functional implications.They show that organellar remodeling is induced by nicotinic stimulation in both non-neuronal and neuronal cells, with the fragmentation of the Golgi apparatus. In neurons, the fragmentation results in the formation of Golgi satellites in dendrites, populated by glycosylation enzymes. In unstimulated neurons, cell surface proteins are produced in the endoplasmic reticulum of neurites and transported to the cell surface with unmodified, high mannose N-glycans. By contrast, the formation of Golgi satellites in dendrites results in proteins carrying complex N-glycans. These complex N-glycans on the nicotinic receptors correlate with a higher activity/response to stimulation.It remains unclear how the vesicles containing glycosylation enzymes are forming Golgi satellites where there was none previously. It is not clear whether the main brake to Golgi satellite formation in unstimulated neurons is the centripetal trafficking of Golgi membranes or if other processes are involved.

We agree it is unclear how the puncta containing Golgi enzymes are forming where none were there previously. We speculate in the paper that the Golgi puncta form through a similar process to Golgi fragmentation induced by microtubule depolymerization during nocodazole treatment. There, Golgi puncta arise in the cell periphery at ER exit sites (ERESs) due to ER-departing Golgi enzymes being unable to traffic back to the Golgi because of the absence of microtubules. Our data suggests some version of this process occurs during neuronal stimulation because the newly formed Golgi puncta (i.e., satellites) arise at ERESs and have a similar size as Golgi puncta formed in nocodazole-treated cells. The exact mechanism remains an open question and is a subject we are now investigating.

This is an interesting study that links multiple processes and proposes an integrated model for how neurons mature in response to stimulation by re-organizing spatially the secretory pathway and glycosylation machinery. It is likely to ignite further work in the domain of neuronal plasticity.The mechanistic insights driving Golgi fragmentation are very limited. The authors are using the term "electrically excited neurons". Is the fragmentation the result of an influx of Ca^2+^ or another process? Is it possible to have some insights from the Hek293 system?

Previous work assaying fragmentation of the Golgi apparatus in the soma of neurons has tied the fragmentation to influx of ca^2+^ during increased neuronal activity (Thayer, Jan et al., 2013). We mention this possibility in the discussion. An additional possibility is that Golgi satellite formation arises from the loss of CAMPSAP2 from the Golgi (known to occur during neuronal activity (Yau, van Beuningen et al., 2014)). CAMPSAP2 recruits GM130 to the Golgi to enable non-centrosomal microtubule growth from the Golgi (Sanders and Kaverina, 2015; Wu, de Heus et al., 2016). If these noncentrosomal microtubules play a role in directing ER-derived transport intermediates toward the Golgi, then their loss from the Golgi during neuronal stimulation might help explain the accumulation of ER-departing Golgi enzymes at ERESs. We are currently investigating these two possibilities using HEK293 cells. To avoid confusion about the meaning of “electrically excited neurons”, we have replaced this phrase with “stimulated neurons” throughout the text.

Since the study is about the formation of Golgi outposts and their population with glycosylation enzymes, I think it would be important to specify the sequences of the reporters used. This is important from a membrane trafficking point of view, especially as the authors show that these outposts are not equivalent to full Golgi (no GM130).For instance, usually, GAL-T reporter is not a full enzyme cloned with a fluorescent protein.

We agree and have now specified the sequences of the reporters used*.*

Also important: there seems to be a mistake in the nature (or naming) of GalNac-T2. This name usually applies to the polypeptide N-acetylGalactosamyl transferase, one of the enzymes that initiate mucin-type O-glycosylation. It does not act on N-glycans and is not a "late-acting enzyme" as mentioned in the text. The N-acetyl-glucosaminyl transferases are the ones acting on high Mannose and usually named GNT-I, II, etc…

We apologize for this confusion. The GalNaC-T2 used in this study is N-acetylGalactosaminyl transferase-2, which is involved in O-linked glycosylation as the reviewer pointed out. It is found both in the medial and trans regions of the Golgi. We never intended to limit the Golgi markers used in this study to enzymes acting on N-linked glycans. Additional details about the enzyme markers used in this study have been added to the text to address this concern.

In addition, the authors should cite other works studying Golgi plasticity and glycosylation in response to external stimuli. For instance, PMCID: PMC3964318: RNAi screening reveals a large signaling network controlling the Golgi apparatus in human cells

We thank the reviewer for raising this point. We have now cited this paper in the discussion.

The authors do not discuss how the cycle stimulation-increase in terminal glycosylation-higher nicotinic receptor activity might result in an amplification loop stabilizing further Golgi satellites.

This is an interesting point. However, as we have no data addressing this possibility we felt it best not to discuss it.

Conversely, are Golgi satellites reversible?

Yes, the formation of Golgi satellites with nicotine exposure is reversible and is described in Figure 1—figure supplements 1 and 2.

Reviewer #2:In this manuscript, Govind and colleagues studied glyco-modification of membrane receptors after neuronal activation. They discovered that Golgi undergoes fragmentation and forms satellite structures in the dendrites after neuronal activation, which act as a micro-secretory system to regulate protein glycosylation. They focused on the nicotinic acetylcholine receptors and showed that the N-glycans on these receptors transformed from an immature, highly mannose form to a mature and complex form containing sialic acid upon neuronal activation. The experiments are very well designed and images are convincing. The conclusions are well supported by their data. It is a pleasure to read this beautiful manuscript. Overall, this is an important and innovative study.There are a few technical issues for the authors to address before publication.1. GalT, St3, Man II and GalNAc-T2 are all late stage glycosylation enzymes. In Figure 1C, many Golgi satellites are St3 positive and GalNAc-T2 negative in HEK cells. However, in Figure 2B, St3 and GalNAc-T2 fully co-localized in the dendrites of neurons. The authors should explain this discrepancy? Is it due to different cell types or distinct mechanisms? The authors should discuss the possibilities. Do other enzymes such as ManII, also co-localize with St3 in HEK cells?

We agree that Golgi satellites forming from ER in dendrites are more homogeneous in their Golgi enzyme composition than the heterogenous Golgi fragments seen in nicotine-treated HEK293 cells., We believe there are two possible explanations for this discrepancy: (1) pre-existing Golgi stacks in the cell body of stimulated neurons undergo differential fission to form Golgi satellites (Quassollo, Wojnacki et al., 2015), or (2) differential rates of cycling of Golgi enzymes back to the ER and entry into ERES-localized Golgi satellites, as occurs during nocodazole treatment (Cole, Ellenberg et al., 1998, Miles, McManus et al., 2001), generates the heterogenous enzyme populations in Golgi satellites. Although we do not show Man II distribution in the HEK 293 cells, we demonstrate in neurons that Golgi puncta/satellites are positive for Man II, GalNac, and St3.

2. The conclusion that complex glycosylation increased a4b2R receptor functionality should be interpreted cautiously, as shown in Figure 7F. However, In Figure 6B, total protein level of a4b2R is apparently reduced after swainsonine treatment. The authors did not provide any evidence to exclude that reduced a4b2R protein caused lower current amplitude in Figure 7F. I suggest that the authors should discuss this caveat. Alternatively, is it possible to knock down some glycosylation enzymes without affecting the a4b2R protein levels and then check if a4b2R receptor functionality is impaired? However, this experiment is not essential for publication. If the authors have this data, it is nice to include.

To rule out the possibility that changes in α4β2R surface levels during swainsonine treatment underlie the effect of swainsonine in blocking the increased α4β2R current response under nicotine treatment, we performed two additional experiments involving biotinylation of α4β2Rs at the cell surface under these conditions. First, we repeated the experiment shown in Figure 6B two more times and performed densitometry on the surface biotinylated α4 subunit bands. The average intensity of the α4 subunits after swainsonine treatment was not significantly different from untreated controls (see Figure 7-figure supplement 2), indicating that the surface α4β2R numbers are not reduced by swainsonine treatment. In the second experiment, we used ^125^I-epibatidine binding to measure the number of surface biotinylated surface α4β2Rs, which is a more quantitative assay. Again, there was no significant difference in the ^125^I-epibatidine binding between untreated and swainsonine-treated samples after nicotine treatment. These new experiments have now been added to supplementary materials (Figure 7—figure supplement 2). From these results we conclude that reduced α4β2R receptor functionality in cells treated with swainsonine plus nicotine is not due to reduction in α4β2R levels at the cell surface. Therefore, nicotine’s ability to increase α4β2R functionality is likely due to the drug’s impact on glycoprotein processing of the receptor.

3. In Figure 5D-E, St3-GFP seems not to co-localize with α4β2R-HA after nicotin treatment. Are the images representative? Moreover, it is interesting to show if upon nocodazole treatment, St3-GFP and α4β2R-HA colocalized, as shown in the case of nicotin treatment (Figure 7A-B). It is also interesting to test whether α4β2R undergo glyco-modification after nocodazole treatment. These experiments would fully address similarity and difference between nicotin treatment and nocodazole treatment. If the authors already have this data, it is nice to include. Otherwise, the authors can publish them in another future study.

In Figure 5D-E, St3-GFP does not co-localize with α4β2R-HA because the anti-HA Ab labeling was performed in a way that only cell-surface α4β2Rs were labeled and not intracellular pools. In this experiment, the anti-HA Ab labeling was only used to show that the neurons were transfected and expressing the α4β2Rs. We have rewritten the text to make this point clearer.

The question about whether nocodazole treatment causes changes to α4β2Rs that are similar to those under nicotine treatment is interesting and we have begun to address this. Nocodazole treatment alone does not change α4β2R glycosylation, nor does it increase ^125^I-epibatidine binding as with nicotine treatment. We have not yet assayed the ligand-evoked current changes. However, nocodazole- and nicotine- treatment together cause an increase ^125^I-epibatidine binding significantly larger than with nicotine alone, which suggests that nocodazole treatment adds to the effects of nicotine. As further work is needed to understand this, we are not including it in the paper.

4. In Figure 7D-E, Biochemical data show that the surface receptors undergo endosomal recycling and glyco-remodelling within 5 hours after nicotin treatment. Can the author also show that in cultured HEK cells or neurons, EEA1 colocalizes with α4β2R-HA as early as 5 h after nicotin treatment?

We have performed this experiment in neurons and indeed find that α4β2R-HA colocalizes with EEA1-labeled endosomes. Importantly, much of the overlap has a slight offset, similar to what we observed in Figure 8E in the manuscript. These findings are consistent with α4β2R-HA being endocytosed, trafficked to endosomes and then to Golgi satellites. We have added this new data to the paper in Figure 7—figure supplement 1.

5. It was reported that golgi outposts rarely distributed in axons of fly sensory neurons. Do Golgi satellites localize in axons?

We do have evidence that Golgi satellites localize in axons. However, we have not included this data in this study as we plan to publish it in a future study.

Reviewer #3:[…]1. Potential overexpression artifacts. A lot of confusion in the field has resulted from the use of over-expressed membrane cargo proteins or organelle markers. As seen with endogenous Golgi proteins including GM130, GRASP55/65, giantin … and GFP-tagged markers such as GalT-GFP or ManII-GFP (after moderate expression), Golgi membranes (i.e Golgi outposts or GO) are rarely found in dendrites (see Reviews). Consistent with this, extensive 3D EM reconstruction of dendrites in hippocampal and cortical pyramidal neurons (e.g., Wu et al., 2017, doi: 10.1073/pnas.1701078114, Cui-Wang et al., doi: 10.1016/j.cell.2011.11.056) have shown a high abundance of dendritic ER, ERGIC and endosomes but a very rare occurrence of stacked membranes characteristic of bona fide Golgi elements in dendrites. As shown for example by Horton and Ehlers with GalT-GFP, co-expression of membrane cargo including VSVG greatly enhances the number of GO in dendrites. In a comparable manner, the data shown here in Figure 2C ii versus 3B-C also indicate that increasing secretory load in the secretory pathway induces the formation/recruitment of ManII-GFP containing elements in dendrites. This issue could be addressed by performing additional control experiments.

We addressed this concern by moderately expressing ManII-GFP, GalNac-T2-mCherry, and ST3-Halo in pairs, or singly staining with internalized fluorescently tagged WGA or a St3 antibody in dendrites (Figure 4—figure supplement 1). In all cases, the overlap between the different markers is in the range of 70-80%. The one exception is the overlap between ManII-GFP and the St3 antibody, where the overlap is ~50%. The lower overlap is consistent with our finding that expressed, tagged St3 only labels 50% of the number of endogenous Golgi satellites stained by the same St3 antibody (Figure 2D and E in the manuscript). As an additional test of whether co-expression of nicotinic receptor subunits with expressed tagged St3 alters the results, we examined how bicuculine treatment altered the number of Golgi satellites measured using the St3 antibody. As displayed in Figure 2—figure supplement 2­, bicuculine treatment resulted in the same increase in the number of St3 antibody-stained puncta as found during nicotine treatment, where co-expression of α4β2Rs is required to observe the increase. These results argue against there being overexpression artifacts in our results.

2. Are Golgi satellites actually Golgi or neuron specific endosomal compartments? The distribution pattern of endogenous St3 shown in Figure 2D strikingly differ from that of other med/late Golgi markers used in the study. In particular, it seems that St3 containing structures (i.e here the endogenous protein) are way more abundant in dendrites than cis/late Golgi membranes labelled with GalT, ManII and GalNacT2. Is this the case? If so, how can the authors rule out that, in neurons, St3 is associated both to the late-Golgi and to non-Golgi membranes such as TGN/endosomal structures or other endomembrane systems, depending on neuronal activity ? Indeed, the number and morphology of St3 containing structures in dendrites is more reminiscent of the endosomal pathway and somehow resemble that of pGolT (Mikhaylova et al., 2016), a synthetic membrane protein containing both TGN/endosomal and Golgi targeting motifs. Rephrasing some statements and performing additional experiments would help addressing this.

The suggested experiments were performed by measuring the overlap between puncta labeled by endogenous and expressed tagged ST3 and puncta labeled by endosomal markers EEA1, VPS35 or internalized transferrin receptor (Figure 4—figure supplement 2). The overlap was in the range of ~25% for puncta labeled by endogenous and expressed tagged ST3 and puncta labeled by ManII-GFP, GalNac-T2-mCherry, or fluorescently tagged WGA. These results support the idea that Golgi satellites are organelles that are distinct from endosomes.

3. Functional relevance of data in HEK cells. Deglycosylation assays shown in Figure 6 and 7 are compelling but, although I may have missed something, were performed in HEK cells only. The notion that neuronal activity reshapes the neuronal surface glycoproteome is thus based primarily on increased levels of surface and internalized WGA binding proteins in neurons after treatment with nicotine or bicuculline. The title of the paper is thus somehow misleading. Experiments on the surface expression, glycosylation and function of endogenous nicotinic receptors in neurons would significantly increase the impact of the study.

We appreciate this concern and now discuss in the revised text why we cannot measure surface expression and glycosylation status of endogenous nicotinic receptors in neurons. Unlike nicotinic receptors at neuromuscular junctions in the periphery, or brain glutamate and GABA receptors, brain nicotinic receptors (such as the α4β2Rs used in our study) are not found clustered at postsynaptic sites at any known synapse. These receptors are also heterogenous in structure, with their subunits often co-assembling with subunits other than α4 and β2 subunits. Their ligand-evoked responses are at best ~10-50-fold smaller than glutamate or GABA_A_ receptor evoked-responses at brain excitatory and inhibitory synapses. Moreover, α4β2Rs are only one of several nicotinic receptor subtypes found in the brain, and only a small percentage of neurons in vivo and in vitro express nicotinic receptors. Indeed, in the neuronal cultures used in our study, we found that only 5-10% of neurons had endogenously expressed α4β2Rs (Govind, Walsh et al., 2012). As discussed below, differences in neuronal excitability increase current response variability. It is no wonder that there are no published reports measuring either endogenous α4β2R expression levels or current responses in neuronal cultures and the few performed in brain slices tend to be contradictory (e.g., Nashmi, Xiao et al., 2007, Baker, Mao et al., 2013). The levels of α4β2Rs appear to be just too small, highly variable and/or in processes far from the soma where currents can be assayed.

In addition to the difficulties assaying endogenous α4β2R expression and functionality in neurons, there are no antibodies available that recognize surface antigens specific to the α4β2Rs. For quantifying α4β2R properties, we thus expressed β2 subunits tagged with HA in their extracellular domain. Using HA specific antibodies, we showed that nicotine exposure altered the surface expression of transfected α4β2Rs in neurons (Figures 5D, E). It also led to α4β2R endocytosis and trafficking into Golgi satellites (Figure 8G). In experiments performed in neurons, we found a significant increase in the surface binding of sialic-acid targeted WGA after neuronal stimulation by bicuculine treatment (Figure 8C). WGA, along with transfected α4β2Rs, were also endocytosed and trafficked into the same Golgi satellites in dendrites of nicotine-stimulated neurons (Figure 8G). This indicated that many endogenous surface glycoproteins (not just α4β2Rs) undergo changes in sialylation with changes in neuronal activity. Complimentary to these results are the findings of Boll et al., (2020), who identified over a hundred synaptic proteins in synaptosomes whose sialic acid content was altered after neuronal stimulation. Of particular interest are the ionotropic glutamate receptors. As identified by Hanus et al., (2016), glycans on cell-surface GluK2- and NR2A- containing receptors are at least partially immature, high-mannose glycans. In Boll et al., (2020), sialic acid was found to be rapidly added to these two receptors in synaptosomes. In other studies, it was found that GluK2-containing kainate receptors in neurons can traffic through Golgi satellites (Evans, Gurung et al., 2017) and that they undergo functional changes caused by changes in their glycosylation (Vernon, Copits et al., 2017). Finally, various other ion channels transporters, receptors, and surface adhesion molecules have been shown to undergo dynamic regulation of their sialylation in neurons (Scott and Panin 2014). These findings suggest that, similar to α4β2Rs, many surface glycoproteins become sialylated during neuronal stimulation, which could result from Golgi satellite formation. We have added this new information to the text to help readers appreciate the functional relevance of our findings.

We believe that the only reliable way to measure α4β2R functional upregulation by nicotine is using heterologous expression of the α4 and β2 subunits in cultured cells such as HEK293 cells. This approach guarantees that only nicotinic receptors containing a known set of subunits (i.e., α4 and β2 subunits) are assayed free of other nicotinic receptor subtypes with relatively little cell-to-cell variation and with a known nicotine concentration during upregulation. The issue of cell-to-cell variation is important because the same neuron cannot be recorded from using electrophysiology during the time required for nicotine exposure. Another advantage of heterologous expression in the HEK293 system is that the cells are not excitable except with the addition of nicotine or other α4β2R agonists. This feature is important because the number of Golgi satellites that α4β2Rs likely traffics through to become functionally upregulated could be increased by other forms of excitability than nicotine in neurons but not in HEK293 cells.

We have considered the possibility of specifically blocking sialic acid addition by knocking-down sialyl-transferases as another test for the functional relevance of sialyation in our system. The problem with sialyl-transferase knock-down, however, is that there are at least four potential sialyl-transferase genes that would be needed to be knocked down to prevent sialic acid addition in cells. Because this has never been done before, another issue is whether it would cause off target effects from other roles played by sialic acid. For these reasons, we relied on the tried and true method of swainsonine treatment to study the effect of blocking sialylation in our system.

Although we were unable to assay endogenous α4β2Rs in neurons for testing how functional upregulation correlates with changes in glycan processing, we did perform experiments assaying how nicotine exposure affects surface expression of transfected α4β2Rs in neurons (Figures 5D, E) and how nicotine exposure affects α4β2R endocytosis and trafficking into Golgi satellites (Figure 8G). To demonstrate that functional upregulation correlates with changes in glycan complex processing of α4β2Rs rather than to changes in surface α4β2R levels in HEK293 cells, we provide new data demonstrating that the number of cell surface α4β2Rs does not change with swainsonine treatment (see Figure 7-figure supplement 2).

Along the same line, including functional data directly addressing whether reported changes in glycosylation impacts neuronal excitability in this context would also enhance the impact of the results. 

We have shown that inhibiting sialylation using swainsonine treatment decreases nicotine-induced membrane currents in HEK293 cells expressing α4β2Rs. Other studies, including Boll et al., (2020), have demonstrated that changes in glycan sialic acid content impacts neuronal excitability (Boll, Jensen et al., 2020). One specific study showed that applying neuraminidase (which cleaves sialic acid off surface glycans) altered the steady-state activation and inactivation of voltage-gated sodium channels in neurons (Isavev et al., 2007), with changes in sodium channel function reducing neuronal excitability and increasing hippocampal seizure threshold, thereby reducing seizures. Another study by Minami et al., (2016) found that a neuraminidase inhibitor reduced LTP at CA3 hippocampal synapses, suggesting that sialic acid addition is needed for LTP (Minami, Saito et al., 2016). The same group found that changes in in vivo neuraminidase activity correlated with previous findings on LTP (Minami, Meguro et al., 2017). In a different study, the same group found that changes in sialic acid content regulates glutamate release from hippocampal neurons via a ca^2+^ signaling modulation (Minami, Ishii et al., 2018). Together, these findings are consistent with changes in sialic acid content on glycoproteins of neuronal membranes and at synapses having a significant impact on neuronal excitability and synaptic transmission. We have added these references to our discussion to help support our speculation that changes in surface receptor glycosylation through Golgi satellite formation impacts neuronal excitability.

References

Baker, L. K., D. Mao, H. Chi, A. P. Govind, Y. F. Vallejo, M. Iacoviello, S. Herrera, J. J. Cortright, W. N. Green, D. S. McGehee and P. Vezina (2013). "Intermittent nicotine exposure upregulates nAChRs in VTA dopamine neurons and sensitises locomotor responding to the drug." Eur J Neurosci 37(6): 1004-1011.

Boll, I., P. Jensen, V. Schwammle and M. R. Larsen (2020). "Depolarization-dependent Induction of Site-specific Changes in Sialylation on N-linked Glycoproteins in Rat Nerve Terminals." Mol Cell Proteomics 19(9): 1418-1435.

Cole, N. B., J. Ellenberg, J. Song, D. DiEuliis and J. Lippincott-Schwartz (1998). "Retrograde transport of Golgi-localized proteins to the ER." J Cell Biol 140(1): 1-15.

Evans, A. J., S. Gurung, K. A. Wilkinson, D. J. Stephens and J. M. Henley (2017). "Assembly, Secretory Pathway Trafficking, and Surface Delivery of Kainate Receptors Is Regulated by Neuronal Activity." Cell Rep 19(12): 2613-2626.

Govind, A. P., H. Walsh and W. N. Green (2012). "Nicotine-induced upregulation of native neuronal nicotinic receptors is caused by multiple mechanisms." J Neurosci 32(6): 2227-2238.

Miles, S., H. McManus, K. E. Forsten and B. Storrie (2001). "Evidence that the entire Golgi apparatus cycles in interphase HeLa cells: sensitivity of Golgi matrix proteins to an ER exit block." J Cell Biol 155(4): 543-555.

Minami, A., A. Ishii, S. Shimba, T. Kano, E. Fujioka, S. Sai, N. Oshio, S. Ishibashi, T. Takahashi, Y. Kurebayashi, H. Kanazawa, N. Yuki, T. Otsubo, K. Ikeda and T. Suzuki (2018). "Down-regulation of glutamate release from hippocampal neurons by sialidase." J Biochem 163(4): 273-280.

Minami, A., Y. Meguro, S. Ishibashi, A. Ishii, M. Shiratori, S. Sai, Y. Horii, H. Shimizu, H. Fukumoto, S. Shimba, R. Taguchi, T. Takahashi, T. Otsubo, K. Ikeda and T. Suzuki (2017). "Rapid regulation of sialidase activity in response to neural activity and sialic acid removal during memory processing in rat hippocampus." J Biol Chem 292(14): 5645-5654.

Minami, A., M. Saito, S. Mamada, D. Ieno, T. Hikita, T. Takahashi, T. Otsubo, K. Ikeda and T. Suzuki (2016). "Role of Sialidase in Long-Term Potentiation at Mossy Fiber-CA3 Synapses and Hippocampus-Dependent Spatial Memory." PLoS One 11(10): e0165257.

Nashmi, R., C. Xiao, P. Deshpande, S. McKinney, S. R. Grady, P. Whiteaker, Q. Huang, T. McClure-Begley, J. M. Lindstrom, C. Labarca, A. C. Collins, M. J. Marks and H. A. Lester (2007). "Chronic nicotine cell specifically upregulates functional alpha 4* nicotinic receptors: basis for both tolerance in midbrain and enhanced long-term potentiation in perforant path." J Neurosci 27(31): 8202-8218.

Quassollo, G., J. Wojnacki, D. A. Salas, L. Gastaldi, M. P. Marzolo, C. Conde, M. Bisbal, A. Couve and A. Caceres (2015). "A RhoA Signaling Pathway Regulates Dendritic Golgi Outpost Formation." Curr Biol 25(8): 971-982.

Sanders, A. A. and I. Kaverina (2015). "Nucleation and Dynamics of Golgi-derived Microtubules." Front Neurosci 9: 431.

Scott, H. and V. M. Panin (2014). "The role of protein N-glycosylation in neural transmission." Glycobiology 24(5): 407-417.

Thayer, D. A., Y. N. Jan and L. Y. Jan (2013). "Increased neuronal activity fragments the Golgi complex." Proc Natl Acad Sci U S A 110(4): 1482-1487.

Vernon, C. G., B. A. Copits, J. R. Stolz, Y. F. Guzman and G. T. Swanson (2017). "N-glycan content modulates kainate receptor functional properties." J Physiol 595(17): 5913-5930.

Wu, J., C. de Heus, Q. Liu, B. P. Bouchet, I. Noordstra, K. Jiang, S. Hua, M. Martin, C. Yang, I. Grigoriev, E. A. Katrukha, A. F. M. Altelaar, C. C. Hoogenraad, R. Z. Qi, J. Klumperman and A. Akhmanova (2016). "Molecular Pathway of Microtubule Organization at the Golgi Apparatus." Dev Cell 39(1): 44-60.

Yau, K. W., S. F. van Beuningen, I. Cunha-Ferreira, B. M. Cloin, E. Y. van Battum, L. Will, P. Schatzle, R. P. Tas, J. van Krugten, E. A. Katrukha, K. Jiang, P. S. Wulf, M. Mikhaylova, M. Harterink, R. J. Pasterkamp, A. Akhmanova, L. C. Kapitein and C. C. Hoogenraad (2014). "Microtubule minus-end binding protein CAMSAP2 controls axon specification and dendrite development." Neuron 82(5): 1058-1073.